# Deficient spermiogenesis in mice lacking *Rlim*

**Feng Wang[1], Maria Gracia Gervasi[2], Ana Bošković[3], Fengyun Sun[3], Vera D Rinaldi[3], Jun Yu[1], Mary C Wallingford[2†], Darya A Tourzani[2], Jesse Mager[2], Lihua Julie Zhu[1,4,5], Oliver J Rando[3], Pablo E Visconti[2], Lara Strittmatter[6], Ingolf Bach[1,4]***

[1]Department of Molecular, Cell and Cancer Biology, University of Massachusetts Medical School, Worcester, United States; [2]Department of Veterinary & Animal Sciences, University of Massachusetts Amherst, Amherst, United States; [3]Department of Biochemistry and Molecular Pharmacology, University of Massachusetts Medical School, Worcester, United States; [4]Program in Molecular Medicine, University of Massachusetts Medical School, Worcester, United States; [5]Program in Bioinformatics and Integrative Biology, University of Massachusetts Medical School, Worcester, United States; [6]Electron Microscopy Core, University of Massachusetts Medical School, Worcester, United States

*For correspondence:
ingolf.bach@umassmed.edu

Present address: †Mother Infant Research Institute, Tufts Medical Center, Boston, United States

Competing interests: The authors declare that no competing interests exist.

**Abstract** The X-linked gene *Rlim* plays major roles in female mouse development and reproduction, where it is crucial for the maintenance of imprinted X chromosome inactivation in extraembryonic tissues of embryos. However, while females carrying a systemic *Rlim* knockout (KO) die around implantation, male *Rlim* KO mice appear healthy and are fertile. Here, we report an important role for *Rlim* in testis where it is highly expressed in post-meiotic round spermatids as well as in Sertoli cells. Systemic deletion of the *Rlim* gene results in lower numbers of mature sperm that contains excess cytoplasm, leading to decreased sperm motility and in vitro fertilization rates. Targeting the conditional *Rlim* cKO specifically to the spermatogenic cell lineage largely recapitulates this phenotype. These results reveal functions of *Rlim* in male reproduction specifically in round spermatids during spermiogenesis.

## Introduction

In testes of adult animals, the differentiation of spermatogonial stem cells during spermatogenesis occurs within seminiferous tubules. Spermiogenesis represents a late stage during spermatogenesis, in which post-meiotic round spermatids differentiate into mature spermatozoa by condensation of the spermatid DNA and formation of the sperm head and tail (*O'Donnell, 2014*; *O'Donnell et al., 2011*). Spermatozoa are then released into the lumen of seminiferous tubules, in a process called spermiation, which involves the remodeling and reduction of cytoplasm (*França et al., 2016*). Even though these processes are crucial for male reproduction, they are poorly understood due to lack of knowledge on spermiation molecular mechanisms.

The ubiquitin proteasome system (UPS) plays important roles in male reproduction (*Richburg et al., 2014*) with a testis-specific version of the proteasome complex (*Kniepert and Groettrup, 2014*). Indeed, the ubiquitination of proteins in cells of the testis is required for functional spermatogenesis including spermiogenesis, and multiple steps during the progression of spermatogonial stem cells to mature spermatozoa critically depend on the UPS (*Richburg et al., 2014*). The UPS pathway is critically dependent upon E3 ubiquitin ligases, which provide substrate specificity by selecting target proteins for ubiquitination (*Metzger et al., 2014*; *Pickart, 2001*).

The X-linked gene *Rlim* (also known as *Rnf12*) encodes a RING H2 type E3 ligase (*Metzger et al., 2014*; *Joazeiro and Weissman, 2000*; *Bach et al., 1999*). While *Rlim* mRNA is widely expressed in many organs and cell types, RLIM protein is more selectively detected (*Bach et al., 1999*; *Ostendorff et al., 2006*). In cells RLIM shuttles between the cytoplasm and nucleus. Nuclear transLocation is regulated by phosphorylation, and in many cell types RLIM is primarily detected in the nucleus (*Jiao et al., 2013*), where it controls not only levels and dynamics of various proteins and protein complexes involved in transcriptional regulation (*Bach et al., 1999*; *Ostendorff et al., 2002*; *Krämer et al., 2003*; *Güngör et al., 2007*; *Gontan et al., 2012*; *Johnsen et al., 2009*; *Her and Chung, 2009*; *Huang et al., 2011*; *Wang et al., 2019*), but also its own expression via autoubiquitination (*Ostendorff et al., 2002*). In female mice, *Rlim* functions as a major epigenetic regulator of nurturing tissues. RLIM promotes the survival of milk-producing alveolar cells in mammary glands of pregnant and lactating females (*Jiao et al., 2012*). Moreover, RLIM is crucial for imprinted X chromosome inactivation (iXCI) (*Shin et al., 2010*; *Wang et al., 2016*; *Gontan et al., 2018*), the epigenetic silencing of one X chromosome in placental trophoblast cells early during female embryogenesis to achieve X dosage compensation (*Payer, 2016*). Indeed, due to inhibited placental trophoblast development, the deletion of a maternally inherited *Rlim* allele results in peri-implantation lethality specifically of females (*Shin et al., 2010*; *Wang et al., 2016*). In contrast, males systemically lacking *Rlim* appear to develop normally, are born at Mendelian ratios and are fertile as adults (*Shin et al., 2010*).

Here, we report high and dynamic *Rlim* mRNA and protein expression in the testis of male mice, where expression of RLIM protein is highly detected in round spermatids during spermiogenesis as well as in Sertoli cells. Indeed, we show that *Rlim* is required for the generation of normal sperm numbers with normal sperm cytoplasmic volume. Even though Sertoli cells are known to regulate cytoplasmic reduction in spermatozoa (*O'Donnell, 2014*; *O'Donnell et al., 2011*), our genetic analyses reveal that this activity is mediated by *Rlim* expressed in the spermatogenic cell lineage. These results assign important functions of *Rlim* during spermiogenesis.

## Results

### Rlim expression in testis is highly regulated

To investigate potential functions of *Rlim* in mice in addition to XCI, we examined mRNA expression in various tissues isolated from adult mice via Northern blots. *Rlim* mRNA was detected in many tissues with highest levels in testis (*Figure 1A*), consistent with published results (*Bach et al., 1999*; *Ostendorff et al., 2000*). However, while the *Rlim*-encoding mRNA in most tissues migrated around 7.5 kb, a variant band at 2.4 kb was detected in testis. Based on published RNA-seq data sets on mouse testes isolated at various post-partum stages (*Margolin et al., 2014*), mapping of reads to the *Rlim* locus revealed relatively homogenously distribution over all exons in 6–20 days post-partum (dpp) animals. However, most of the reads in sexually mature (38 dpp) animals mapped in exonic regions upstream of the TGA Stop codon, encompassing the 5' non-coding region and the entire open-reading frame (ORF), while most of the 3' noncoding region was underrepresented (*Figure 1B*). Indeed, consistent with the length of the observed variant *Rlim* mRNA, closer examination the *Rlim* cDNA sequence revealed a consensus alternative polyadenylation site (*Proudfoot, 2011*) starting 69 bp downstream of the TGA Stop codon (*Figure 1C*). Thus, in mature mouse testes a short, variant *Rlim* RNA is generated by alternative polyadenylation (*Tian and Manley, 2017*).

Because expression of *Rlim* mRNA and protein can be strikingly different (*Ostendorff et al., 2006*), we examined RLIM protein in testes using an established RLIM antibody (*Ostendorff et al., 2006*; *Ostendorff et al., 2002*). Consistent with our mRNA analyses (*Figure 1*), western blots on protein extracts of testis, brain, and spleen confirmed high expression of full-length RLIM protein in testis (*Figure 2—figure supplement 1A*). Using immunohistochemistry (IHC) on testes sections, we detected strong immunoreactivity in specific regions of some seminiferous tubules, representing differentiating spermatogenic cells (*Figure 2A*). Moreover, we detected single RLIM-positive cells at the periphery of all tubules. Consistent with published results (*Wang et al., 2016*), little to no RLIM staining was detected in males carrying a Sox2-Cre (*Hayashi et al., 2002*) – mediated conditional knockout of the *Rlim* gene (cKO/Y^Sox2-Cre), as these animals lack RLIM in somatic tissues as well as

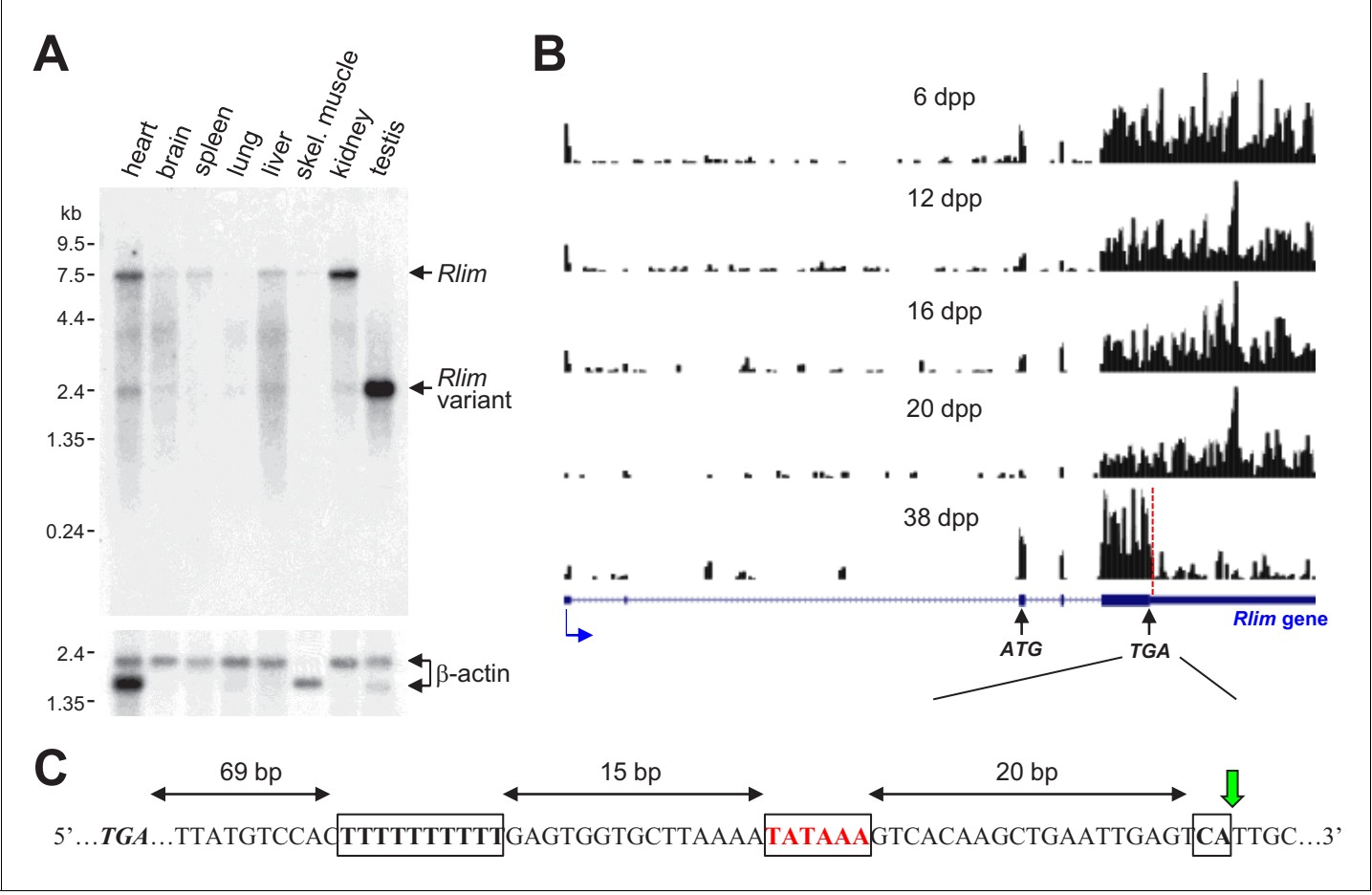

**Figure 1.** A short Rlim mRNA variant highly expressed in testis is generated via alternative polyadenylation in mature male mice. (**A**) A Northern blot containing RNA extracts from various adult mouse tissues (WT for *Rlim*) was hybridized with an *Rlim* probe (upper panel) and a probe recognizing β-actin as loading control (lower panel). (**B**) Modified from the UCSC Genome Browser: Cumulative raw reads from RNA-seq datasets of testes RNA isolated from post-natal mice at 6, 12, 16, 20, and 38 days post-partum (dpp) (*Margolin et al., 2014*) were mapped on the *Rlim* locus (variable scales). Structure of the *Rlim* gene is shown below in blue with boxed exon regions. Protein coding regions are indicated in thicker stroke. Blue arrow indicates direction of transcription. ATG start codon, TGA stop codon and site of alternative polyadenylation sequence (red dotted line) is indicated. Note low relative read density 3' of the alternative polyadenylation site specifically in 38 dpp animals. (**C**) Nucleotide sequence containing an alternative polyadenylation site downstream of the TGA stop codon. Conserved motifs including a T-rich sequence, A/TATAAA, and CA motifs are boxed. The cleavage position is indicated (green arrow).

the germline (*Wang et al., 2016*; *Shin et al., 2014*). Lack of RLIM protein in testes of cKO/Y^Sox2-Cre animals was confirmed by western blot (*Figure 2—figure supplement 1B*). Because high RLIM levels appeared to be expressed at specific stages during spermatogenesis (*Figure 2A*), we performed co-staining using an antibody against peanut agglutinin (PNA) that stains acrosomal structures allowing staging of spermatogenic cells within seminiferous tubules (*Kotaja et al., 2004*; *Oakberg, 1956a*; *Oakberg, 1956b*). Around seminiferous stage II / III, RLIM levels are detectable but low in all spermatogenic cells, including post-meiotic step 2/3 spermatids (*Figure 2B*). However, indicated by RLIM/PNA co-staining, RLIM levels are high in step 6–8 round spermatids (stages VI–VIII), a mid-timepoint in spermiogenesis before spermatids begin to elongate (*O'Donnell, 2014*; *Qian et al., 2014*). Thus, RLIM protein levels are low in step 1–5 spermatids, dramatically upregulated in step 6–8 spermatids, and then downregulated in elongating spermatids at step 10 prior to their release (*O'Donnell et al., 2011*; *Figure 2B*). We noted that the nuclei of RLIM-positive cells located at the periphery of seminiferous tubules displayed a triangular shape characteristic for Sertoli cells (*Figure 2—figure supplement 1C*). To identify this RLIM-positive cell type (*Figure 2A*), we performed IHC, co-staining with antibodies against RLIM and the Sertoli cell marker GATA1 (*Yomogida et al.,*

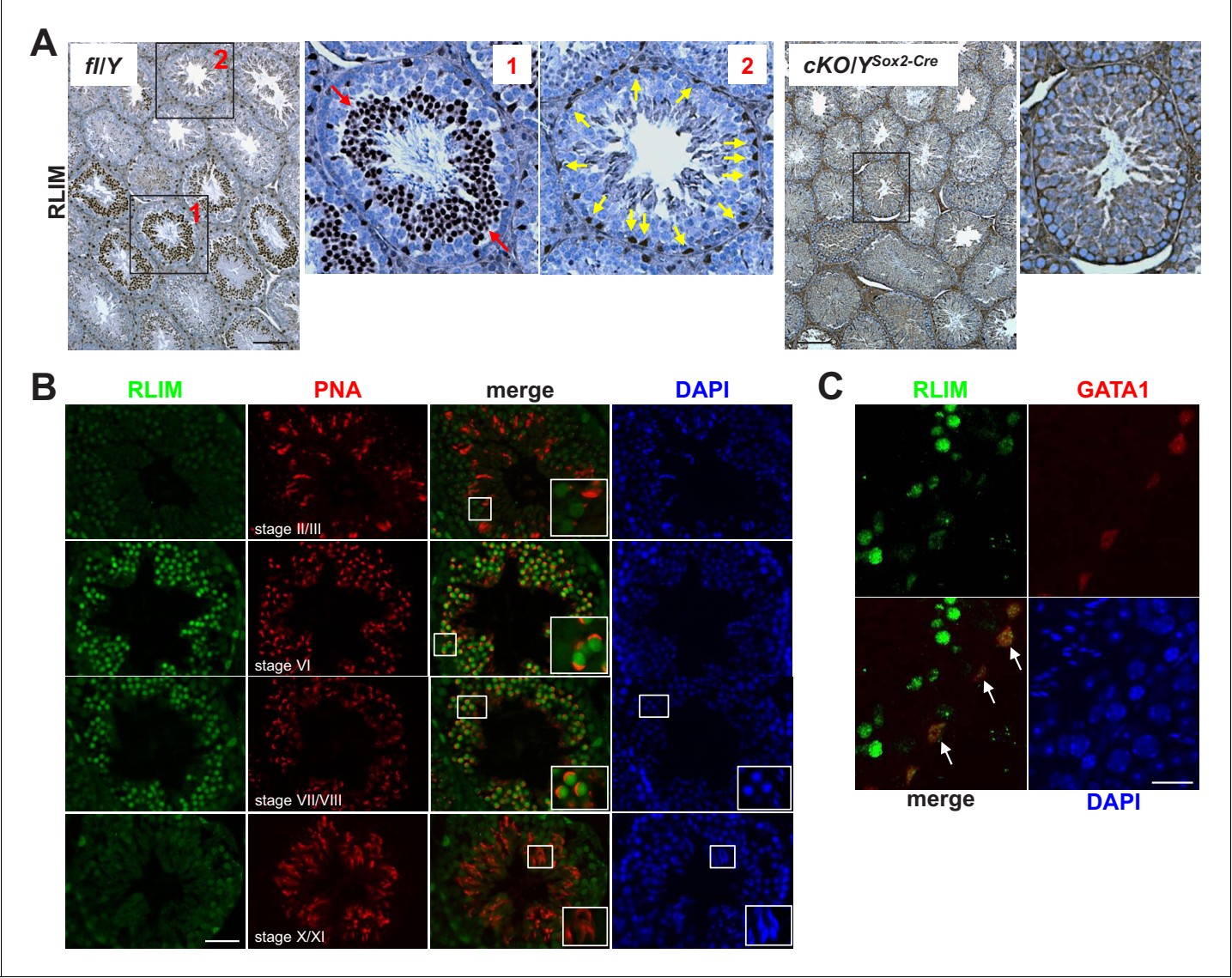

**Figure 2.** High RLIM protein expression specifically in round spermatids and in Sertoli cells. Tissue sections of mouse testes were stained using IHC with indicated antibodies. Boxed areas are shown in higher magnification. (A) DAB staining of testes sections generated from fl/Y and, as negative control, *Rlim* cKO/Y$^{Sox2-Cre}$ males littermates using antibodies against RLIM. Left panel: fl/Y male. Red arrows (box 1) and yellow arrows (box 2) point at spermatogenic cells and cells located in the periphery of seminiferous tubules that exhibit high RLIM staining, respectively. Right panels: cKO/Y male. Scale bars = 150 μm. (B) IHC on fl/Y testis co-staining with antibodies against RLIM (green) and PNA (red) to determine differentiation stages of spermatogenic cells within seminiferous tubules (indicated). Scale bar = 60 μm. (C) IHC on fl/Y testis co-staining with antibodies against RLIM (green) and GATA1 (red), a Sertoli cell marker. Scale bar = 25 μm.

The online version of this article includes the following figure supplement(s) for figure 2:

**Figure supplement 1.** RLIM protein in testis.

1994). Positive co-staining identified these cells as Sertoli cells (*Figure 2C*). This dynamic and regulated expression of *Rlim* suggests roles during male reproduction.

## Diminished production and functionality of Rlim KO sperm

Next, we investigated functions of the *Rlim* gene in males using our conditional knockout (cKO) mouse model (*Shin et al., 2010*). Results on male mice systemically lacking *Rlim* induced by Cre recombinase transgenes (*Shin et al., 2010*; *Wang et al., 2016*; *Shin et al., 2014*) or by germline KO (*Figure 3—figure supplement 1A*) reveals that *Rlim* has no essential functions during male

embryogenesis and post-natal development. The germline KO/Y animals were generated by crossing heterozygous *Rlim* KO females (fl$_m$/KO$_p$) with WT/Y males (*Shin et al., 2010*; *Shin et al., 2014*). To explore potential roles of the *Rlim* gene during male reproduction, we compared testes of 8-week-old animals systemically lacking *Rlim* either by a germline *Rlim* KO or with a Sox2-Cre-mediated *Rlim* cKO (cKO/Y$^{Sox2-Cre}$) with fl/Y littermate controls. Indeed, in cKO/Y$^{Sox2-Cre}$ and KO/Y mice the size and weight of the testis was significantly decreased when compared to their respective fl/Y littermates, (*Figure 3A,B*; *Figure 3—figure supplement 1B*). This was accompanied by lower numbers of mature sperm isolated in Caudal swim-out experiments (*Figure 3C*; *Figure 3—figure supplement 1C*). Even though these numbers are biased towards motile sperm, these data suggest diminished sperm production. To examine plausible causes for these phenotypes, we analyzed PAS-stained testis sections. However, we did not detect obvious abnormalities of specific testicular cell types during spermiogenesis in animals lacking *Rlim*, and sperm release into tubules appeared normal (*Figure 3—figure supplement 1D*, not shown). Moreover, in IHC staining testes sections with antibodies against cleaved caspase 3, there were no signs of increased apoptosis in the spermatogenic cells lacking *Rlim* (not shown). Examining epididymal epithelia, which also express RLIM (*Figure 3—figure supplement 2A*), revealed no significant differences in weight between fl/Y and cKO/Y$^{Sox2-Cre}$ animals (*Figure 3—figure supplement 2B*). Visualizing tubules that form the inner epididymal layers via de-lipidation (*Sylwestrak et al., 2016*; *Tomer et al., 2014*), we did not detect major differences between genotypes (*Figure 3—figure supplement 2C*). Our results indicate testicular phenotypes in mice lacking *Rlim*.

As *Rlim* is a transcriptional regulator and the structural integrity of testes appeared to be generally intact in cKO/Y$^{Sox2-Cre}$ males (*Figure 2A*), we next examined effects of *Rlim* on genome wide gene expression. Thus, we performed RNA-seq experiments on RNA isolated from total testis of 8-week-old animals, comparing global gene expression in cKO/Y$^{Sox2-Cre}$ and fl/Y male littermates. These experiments including library construction, sequencing, and data processing were performed as previously described (*Wang et al., 2017*). Consistent with findings that attribute both positive and negative functions of RLIM for gene transcription (*Bach et al., 1999*; *Gontan et al., 2012*), statistical analyses revealed 118 down-regulated and 83 up-regulated genes (threshold p<0.05) in cKO/Y$^{Sox2-Cre}$ animals (*Figure 3C,D*). Gene ontology analyses revealed that functions of these genes fell mostly in eight categories with around half of all differentially expressed genes involved in signaling (27.5%) and regulation of metabolism (22.5%), in particular regulatory functions on lipid metabolism (*Figure 3E*). Other genes are involved in transcription/chromatin (14%), cell organization (6%), transport (6%) or the UPS (3%), while 9.5% of gene functions occupy multiple other cellular pathways. Around 10.5% of transcripts constituted non-coding (nc) RNAs. No major differences in functions between up- and down-regulated genes were detected in cKO/Y$^{Sox2-Cre}$ testes. Combined, these results suggest functions of *Rlim* during spermatogenesis.

Next, we investigated the characteristics and functionality of sperm isolated from the Cauda of 8-week-old males via swim-out. Indeed, cKO/Y$^{Sox2-Cre}$ sperm displayed elevated rates of morphological abnormalities, including coiled midpieces and head malformations (*Figure 4A,B*). However, the sperm capacitation-induced phosphorylation pathways, acrosomal status and induced acrosome reaction in *Rlim* KO sperm were similar to controls as judged by western blot and PNA staining, respectively (*Figure 4—figure supplement 1A–E*), indicating that the deletion of *Rlim* did not affect general signaling. To examine sperm motility, we used CEROS computer-assisted semen analysis (CASA) in swim-out experiments at T0 and after 60 min (T60) under conditions that allow capacitation comparing cKO/Y$^{Sox2-Cre}$ males with fl/Y littermates. The results revealed significant motility deficiencies of cKO/Y$^{Sox2-Cre}$ sperm. Indeed, cKO/Y$^{Sox2-Cre}$ sperm displayed decreased total motility, and out of the total motile sperm the percentages of sperm with progressive motility was lower, while the percentages of slow and weakly motile sperm populations were higher (*Figure 4C–E*). Thus, cKO/Y$^{Sox2-Cre}$ sperm is less motile. Next, we tested for possible functional consequences of these deficiencies during in vitro fertilization (IVF) (*Sharma et al., 2016*), using oocytes originating from WT females and sperm isolated from either fl/Y or cKO/Y$^{Sox2-Cre}$ littermates. In these experiments, the numbers of oocytes and sperm cells were adjusted to 100–150 and 100,000, respectively, to achieve fertilization rates of around 80% for the control samples as judged by the number of embryos reaching cleavage stage 24 hr after adding sperm (*Figure 4F,G*). The development of IVF embryos was monitored up to (96 hr) at which point blastocyst stage was reached by the majority of embryos generated by control sperm. Indeed, sperm isolated from fl/Y control animals yielded in

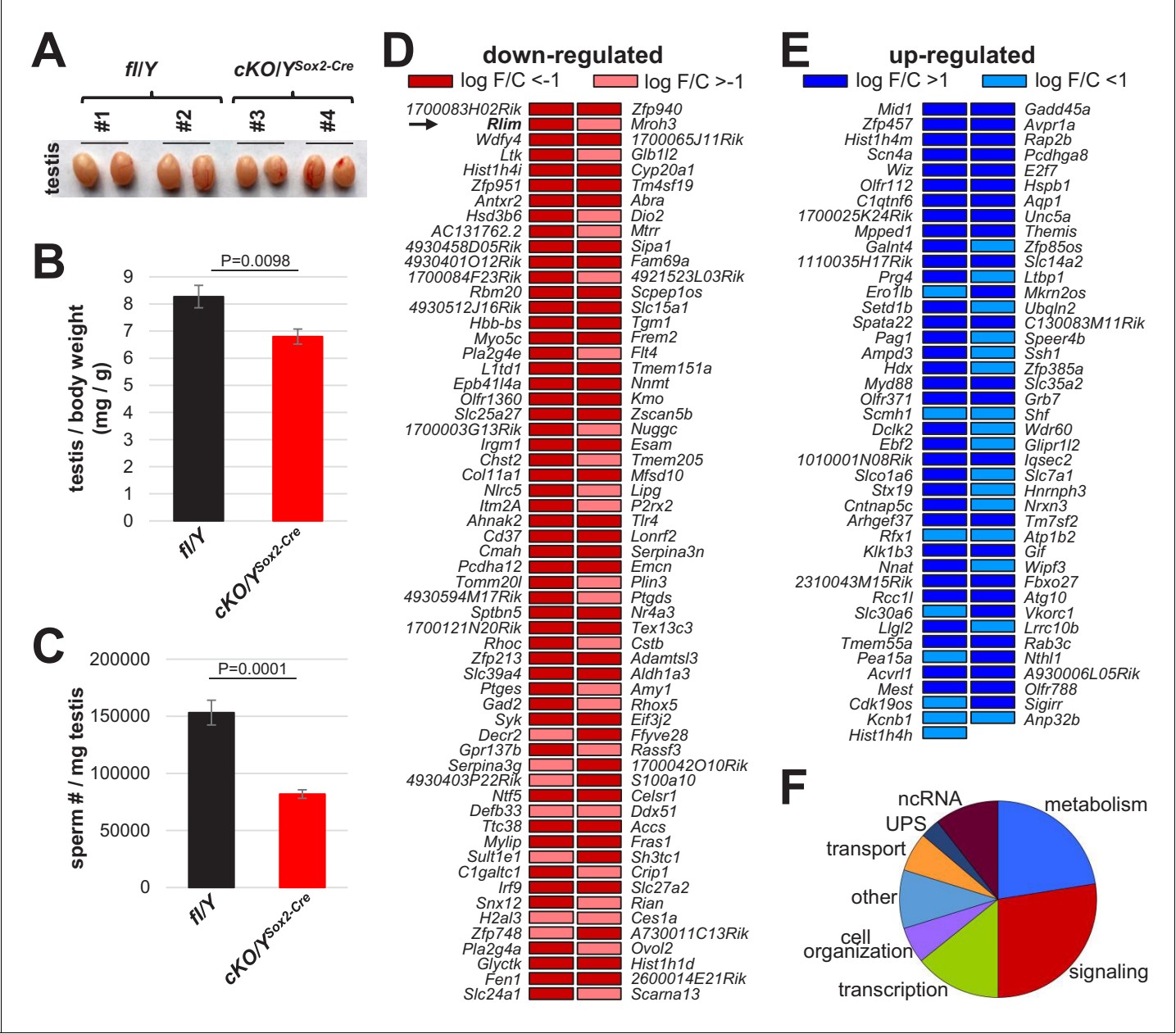

**Figure 3.** Lack of *Rlim* affects sperm production. Males systemically lacking *Rlim* were generated via Sox2-Cre mediated *Rlim* deletion (cKO/Y^Sox2-Cre) and directly compared to fl/Y male littermates at 8 weeks of age. (A) Deletion of *Rlim* results in smaller testes. Representative testes isolated from adult male fl/Y control animals (#1,2) and cKO/Y^Sox2-Cre littermates (#3,4) are shown. (B) Significantly decreased weight of testes isolated from cKO/Y^Sox2-Cre animals (n = 9) when compared to fl/Y littermates (n = 7). Values were normalized against total body weight and represent the mean ± s.e.m. p Values are shown (students t-test). (C) Significantly decreased numbers of sperm in animals lacking *Rlim.* Cauda epididymal sperm were collected via swim-out in HTF medium. After 10 min of swim-out, total sperm numbers were determined (n = 7 fl/Y; n = 9 cKO/Y^Sox2-Cre). s.e.m. and p Values are indicated. (D, E) Differentially expressed genes in testes of fl/Y and cKO/Y^Sox2-Cre mice as determined by RNA-seq experiments on biological replicates. Genes significantly (p<0.05) down-regulated and up-regulated upon the *Rlim* deletion in each experiment are shown in (C) and (D), respectively. Arrow indicates *Rlim.* (F) Differentially expressed genes distribute in eight functional categories that include metabolism, signaling, transcription, cell organization, transport, UPS, ncRNA and other.

The online version of this article includes the following figure supplement(s) for figure 3:

**Figure supplement 1.** Decreased sperm production in males lacking Rlim.

**Figure supplement 2.** Normal epididymal appearance in males lacking Rlim.

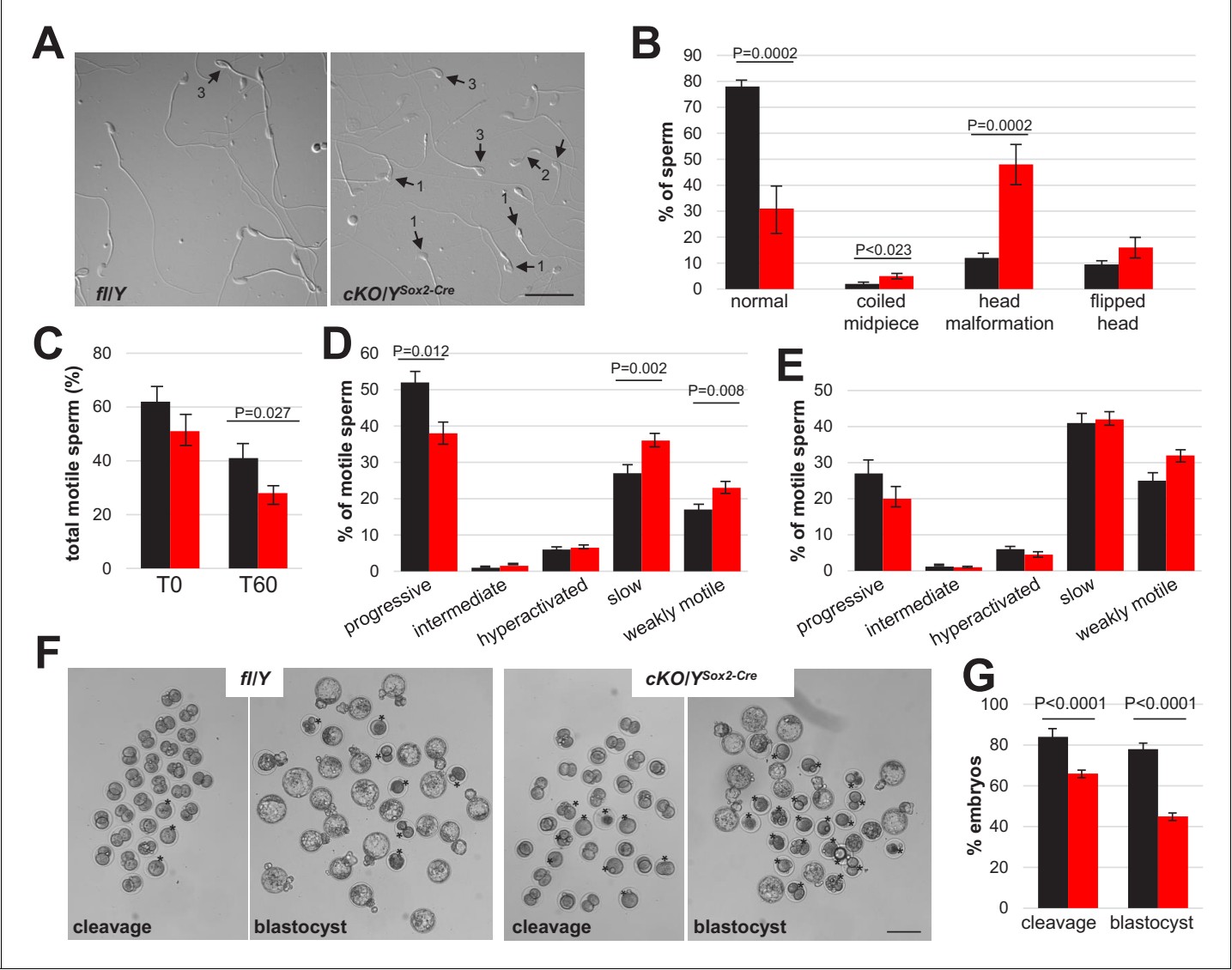

**Figure 4.** Increased abnormalities and decreased functionality in sperm lacking *Rlim*. Cauda epididymal sperm were collected via swim-out from 8-week-old males (n = 7 fl/Y ■; n = 9 cKO/Y$^{Sox2-Cre}$ ■). (A) Sperm morphology was assessed by light microscopy. Representative images of difference interference contrast (DIC) display the morphological patterns found indicated by arrows: 1, Head malformation; 2, Coiled midpiece; 3, Flipped head. Scale bar = 25 µm. (B) Quantification of the morphology patterns. Percentages of normal sperm, coiled midpiece, head malformation, and flipped head out of the sperm population. At least 100 sperm per sample were counted, total sperm counted 1146 for fl/Y and 1077 for cKO/Y$^{Sox2-Cre}$. Values represent the mean ± s.e.m. (C) Sperm motility was evaluated in the swim-out (T0) and after 60 min of incubation in conditions that support capacitation (T60). Sperm motility was examined using the CEROS computer-assisted semen analysis (CASA) system. Percentage of total motile sperm in the population. n = 12, values represent the mean ± s.e.m. (D) Classification of type of motility (progressive, intermediate, hyperactivated, slow, and weakly motile; in percentage) out of the total motile population at T0. n = 12, values represent the mean ± s.e.m. (E) Classification of type of motility (in percentage) out of the total motile population after 60 min of incubation in capacitation conditions (T60). n = 12, values represent the mean ± s.e.m. (F) Representative images for cleavage and blastocyst stages at 24 hr and 96 hr, respectively. Asterisks indicate embryos not reaching anticipated embryonic stage. Scale bar = 100 µm. (G) Summary of IVF results. n = 142 and 313 presumed oocytes for fl/Y and cKO/Y sperm, 7 and 8 animals, respectively. Values represent the mean ± s.e.m.

The online version of this article includes the following figure supplement(s) for figure 4:

**Figure supplement 1.** Evaluation of molecular and physiological events related to sperm capacitation.

84.3% cleavage stage and 77% blastocyst stage embryos, compared to 67.0% and 46.8% for cKO/Y$^{Sox2-Cre}$ sperm, respectively (*Figure 4F,G*). Moreover, the rate of blastocysts per cleavage staged embryos was 92% of for fl/Y sperm and only 70% for cKO/Y$^{Sox2-Cre}$ (*Figure 4—figure supplement 1F*), suggesting that lack of *Rlim* negatively affects embryonic development under in vitro conditions. Thus, sperm isolated from cKO/Y$^{Sox2-Cre}$ animals yielded in significantly less embryos reaching the appropriate developmental stage when compared to fl/Y littermate controls, both for cleavage and blastocyst stages (*Figure 4F,G*). Combined, these data indicate that deletion of *Rlim* in male mice results in reduced sperm functionality.

## Increased size of cytoplasmic droplets in Rlim cKO sperm

Because of the decreased motility of *Rlim* cKO sperm (*Figure 4C–E*), we examined the energetic status including amino acids, glycolysis, TCA cycle, pentose phosphate pathway, and nucleotide biosynthesis via metabolomic profiling of polar metabolites. We isolated 40–45 Mio cauda swim-out sperm each of 3 fl/Y and 5 cKO/Y$^{Sox2-Cre}$ animals, and polar metabolites were measured using liquid chromatography coupled with mass spectrometry (LC-MS). Results showed that, unexpectedly, 66 metabolites were significantly more abundant in the cKO/Y$^{Sox2-Cre}$ sperm, in particular levels of many amino acids, in contrast to only five metabolites that displayed lower levels (*Figure 5A*). Analyses of these diverse metabolites did not yield in the identification of a specific energy production pathway but rather revealed that metabolite intermediates of many cellular pathways are increased in the *Rlim* cKO sperm, suggesting a more general problem. Thus, we included electron microscopy (EM) in our analyses. Interrogating sperm via scanning EM (SEM) revealed significantly more *Rlim* cKO/Y$^{Sox2-Cre}$ sperm displaying cytoplasmic droplets in the midpiece, and these droplets were also increased in size as measured using ImageJ software (*Figure 5B–D*). Because Cauda sperm has matured a considerable time in the epididymis that also expresses *Rlim* (*Figure 3—figure supplement 2A*), to distinguish testicular versus epididymal functions of *Rlim* in droplet formation we extended these studies to testicular sperm. Indeed, isolating testicular sperm from 30 fl/Y and cKO/Y$^{Sox2-Cre}$ animals each, the numbers of isolated sperm per *Rlim* cKO testis were lower but this was no longer significant when compared to controls (*Figure 6A*). We noted that the midpieces from cKO sperm were highly vulnerable to rupturing, while the prevalence of coiled midpieces appeared similar between cKO and control sperm (*Figure 6B*). Moreover, we detected a low number of sperm that exhibited duplicated axonemes specifically in *Rlim* cKO/Y$^{Sox2-Cre}$ but not in fl/Y sperm (*Figure 6B*; *Figure 6—figure supplement 1A*). Again, the sizes of cytoplasmic droplets in testicular sperm were significantly increased (*Figure 6C,D*). Transmission EM (TEM) on testes sections confirmed the occurrence of duplicated axonemes in cKO/Y$^{Sox2-Cre}$ sperm as well as head malformations, while the overall structural integrity of the sperm tail appeared normal (*Figure 6—figure supplement 1A–D*), and no signs of decreased chromatin packaging in sperm heads was detected as judged by sperm head density. Interestingly, the cytoplasmic pockets in the *Rlim* cKO sperm heads appeared more pronounced in sperm of the epididymal Caput region (*Figure 6E,F*). These data reveal excessive cytoplasm in sperm heads and midpieces in males lacking *Rlim* and are consistent with testicular as opposed to epididymal functions of *Rlim*.

## Functions of Rlim during spermiogenesis specifically in the spermatogenic cell lineage

Next, we addressed the question as to the cell type of *Rlim* action during spermiogenesis as RLIM protein is highly detectable both in the spermatogenic cell lineage specifically in round spermatids and in Sertoli cells (*Figure 2*). Because Sertoli cells play major roles in the regulation of spermiogenesis/spermiation (*O'Donnell, 2014*; *O'Donnell et al., 2011*; *França et al., 2016*) and cell numbers are correlated with sperm production capacity (*Griswold, 1995*), we compared number of cells positive for Sertoli cell marker GATA4, as in adult mice GATA4 expression does not vary with the cycle of the seminiferous epithelium (*Yomogida et al., 1994*). Counting GATA4-positive cells within seminiferous tubules, IHC revealed similar numbers of Sertoli cells in testes with or without *Rlim* (*Figure 7—figure supplement 1A,B*), indicating that *Rlim* is not required for Sertoli cell development and differentiation. Moreover, analyzing specific Sertoli cell structures involved in spermiation in testis sections via TEM, we did not detect major structural deficiencies in cells lacking *Rlim* including

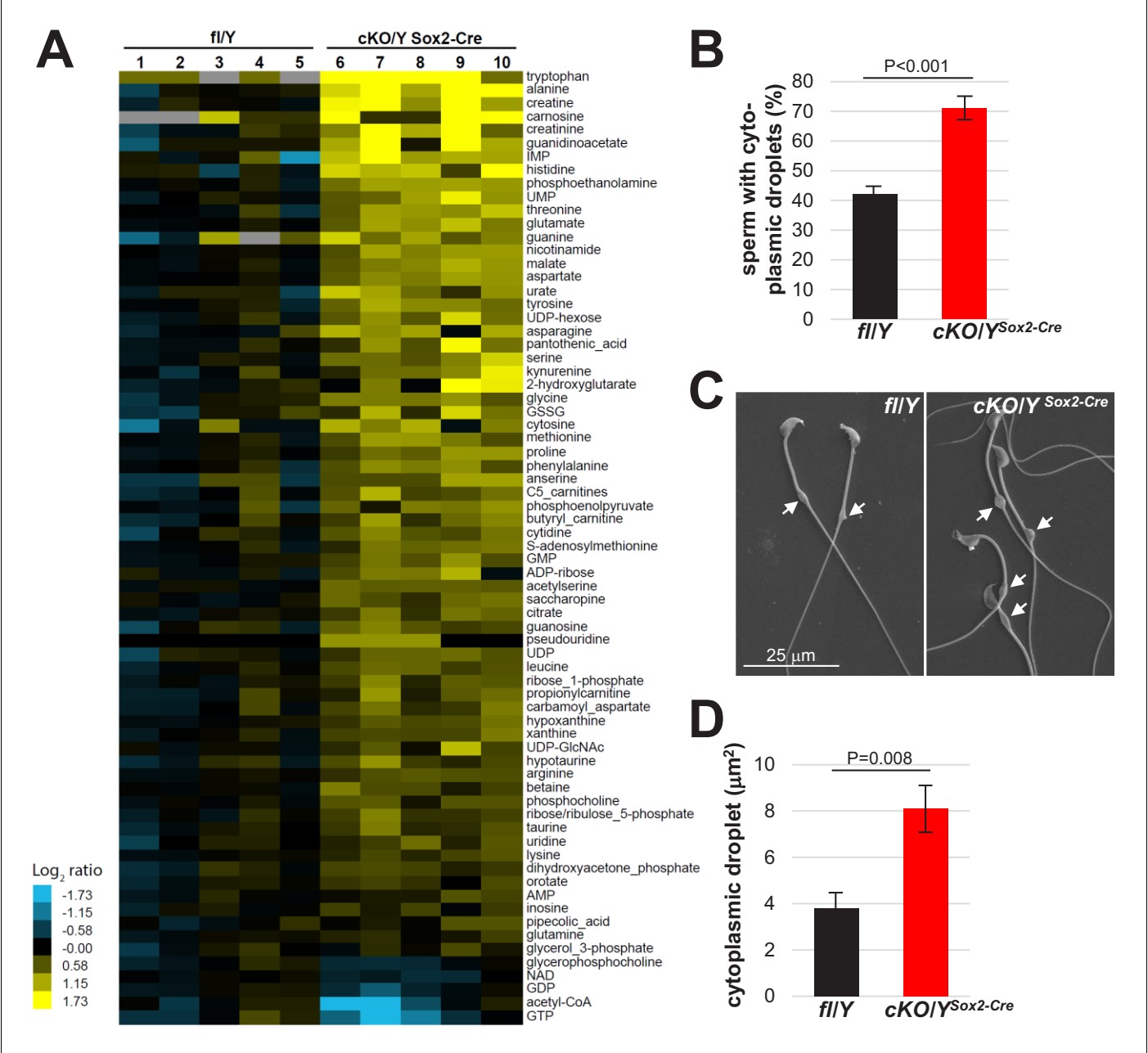

**Figure 5.** Increased size of cytoplasmic droplet in caudal sperm of males lacking *Rlim*. (A) Cauda epidymal sperm were collected from 8-week-old fl/Y or cKO/Y[Sox2-Cre] mice and polar metabolites were determined via LC-MS. Samples were run and data analyzed by the Metabolite Profiling Core Facility at the Whitehead Institute. Note general increased content of metabolites in cKO/Y[Sox2-Cre] sperm. (B) Cauda sperm was collected from 8-week-old fl/Y or cKO/Y[Sox2-Cre] mice and after 10 min separation, immediately fixed in 2.5% glutaraldehyde followed by SEM analysis. Sperm with or without cytoplasmic droplets were counted. n = 250, each. (C) Increased size of cytoplasmic droplets in cKO/Y[Sox2-Cre] sperm. Representative images are shown. Droplets are indicated by arrows. (D) Increased size of cytoplasmic droplets in cKO/Y[Sox2-Cre] sperm. Droplet surface size was determined via ImageJ. n = 100, each.

the formation of apical ectoplasmic specialization (ES) or the apical tubulobulbar complex (TBC), and no obvious signs of defective cytoplasmic reduction (*Figure 7—figure supplement 1C,D*).

In order to genetically elucidate the cell identity of *Rlim* function, we targeted the *Rlim* cKO via *Ngn3-Cre* (also known as *Neurog3-Cre*) (*Schonhoff et al., 2004*) to the spermatogenic cell lineage (*Yoshida et al., 2004*) and via *Sf1-Cre* (*Dhillon et al., 2006*) to Sertoli cells (*Kim et al., 2007*). IHC on testis sections using RLIM antibodies confirmed the correct, specific and penetrant targeting of

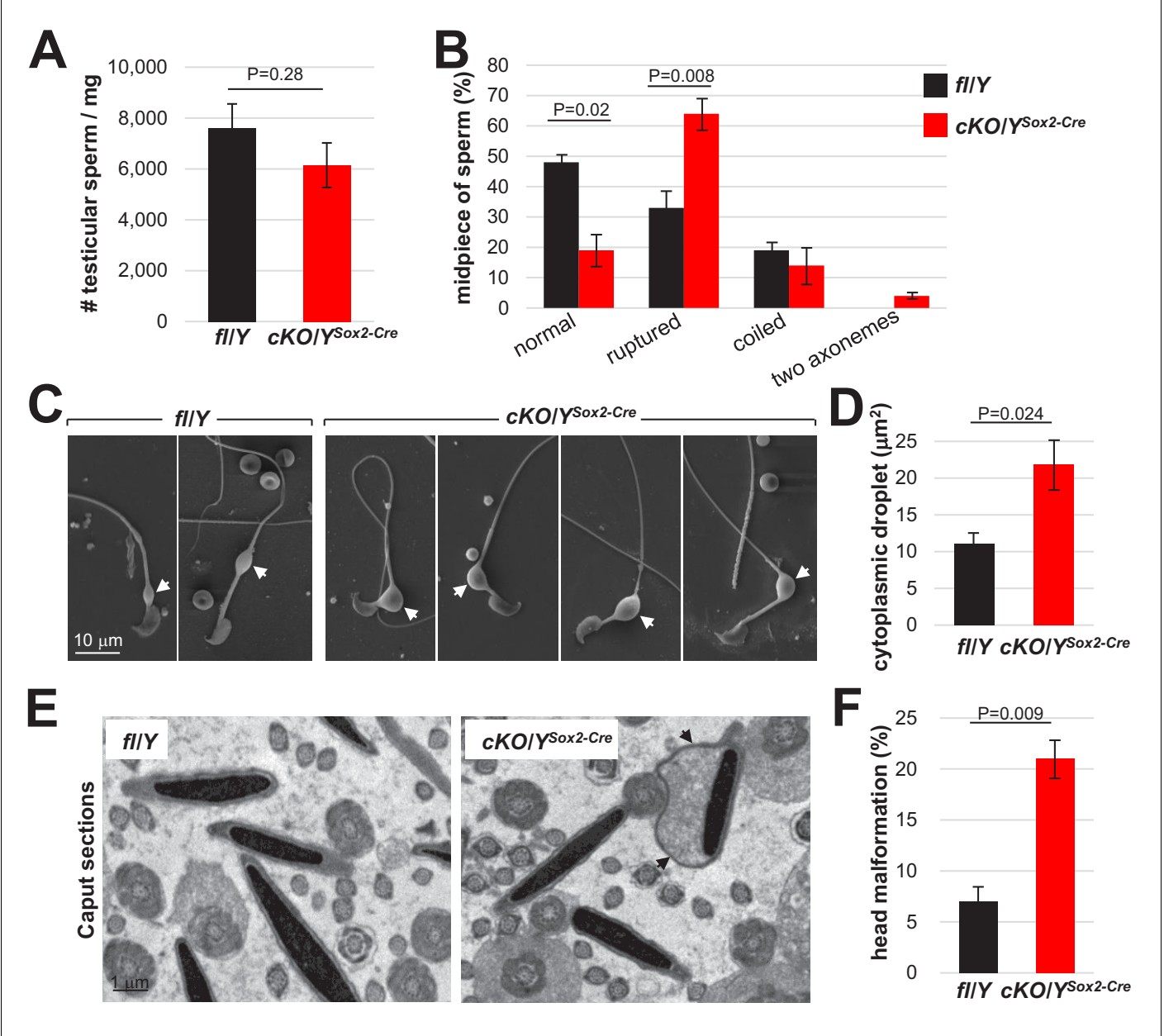

**Figure 6.** *Rlim* plays important roles for cytoplasmic reduction. Analyses of testicular sperm isolated from 8 weeks-old fl/Y or cKO/Y$^{Sox2-Cre}$ mice. (**A**) Quantification of sperm yield normalized against testis weight. n = 30, each genotype. (**B**) Quantification of sperm morphology scoring ruptured, coiled sperm and sperm with two axonemes. n = 250. (**C**) Larger cytoplasmic droplets in cKO/Y$^{Sox2-Cre}$ sperm. Representative images are shown. Droplets are indicated by arrows. (**D**) Increased size of cytoplasmic droplets in cKO/Y$^{Sox2-Cre}$ sperm. Droplet surface size was determined via ImageJ. n = 100, each. (**E**) Sperm head malformations within the epididymal Caput region as determined via TEM. Representative images are shown. Arrows point at cytoplasmic pocket. (**F**) Quantification of sperm exhibiting excessive head cytoplasmic pocket.

The online version of this article includes the following figure supplement(s) for figure 6:

**Figure supplement 1.** Evaluation of sperm structure within testes of cKO/Y$^{Sox2-Cre}$ animals.

both Cre drivers (**Figure 7A**). Because of a mixed background of these Cre-driver mouse lines, cKO males were compared to their respective littermate controls. Indeed, *Rlim* cKO/Y$^{Ngn3-Cre}$ mice displayed decreased testis weights and numbers of mature sperm isolated in caudal swim-out experiments (**Figure 7B,C**, respectively). In contrast, no significant effects on testes weights and sperm numbers were measured in cKO/Y$^{Sf1-Cre}$ animals. Consistent with these findings, SEM analyses of Caudal swim-out sperm revealed significantly increased numbers of sperm containing cytoplasmic

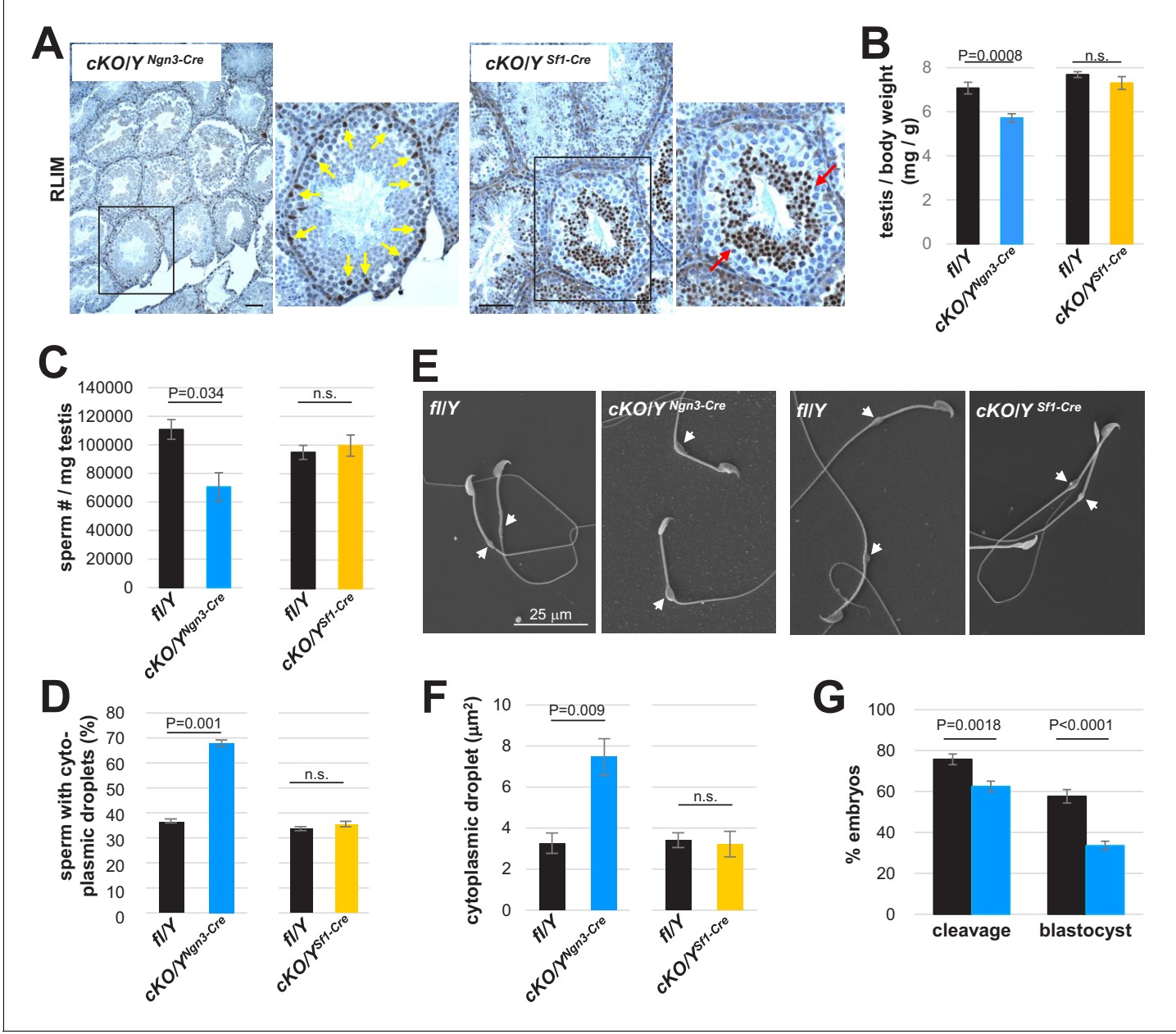

**Figure 7.** Functions of *Rlim* predominantly in the spermatogenic cell lineage. Animals with an *Rlim* cKO in the spermatogenic cell lineage or in Sertoli cells were generated via Ngn3-Cre and SF1-Cre (cKO/Y$^{Ngn3-Cre}$ and cKO/Y$^{SF1-Cre}$), respectively. (**A**) IHC on testis sections of cKO/Y$^{Ngn3-Cre}$ and cKO/Y$^{SF1-Cre}$ animals using RLIM antibodies. Correct targeting is indicated by lack of RLIM specifically in spermatogenic cells but not Sertoli cells (yellow arrows) in cKO/Y$^{Ngn3-Cre}$ animals, and in Sertoli cells but not round spermatids (red arrows) in cKO/Y$^{SF1-Cre}$ males. Scale bars = 75 µm. (**B**) Significantly decreased weight of testes isolated from cKO/Y$^{Ngn3-Cre}$ males but not from cKO/Y$^{SF1-Cre}$ animals (n = 18 fl/Y; 14 cKO/Y$^{Ngn3-Cre}$) (n = 16 fl/Y; 10 cKO/Y$^{SF1-Cre}$). cKO animals were directly compared to their respective fl/Y male littermates at 8 weeks of age. Values were normalized against total body weight and represent the mean ± s.e.m. p Values are shown (students t-test). (**C**) Significantly decreased numbers of sperm isolated from cKO/Y$^{Ngn3-Cre}$ males but not from cKO/Y$^{SF1-Cre}$ animals. Cauda epididymal sperm were collected via swim-out in HTF medium. After 10 min of swim-out, total sperm numbers were determined (n = 7 fl/Y; n = 9 cKO/Y$^{Ngn3-Cre}$; n = 9 fl/Y; n = 11 cKO/Y$^{SF1-Cre}$). s.e.m. and p values are indicated. (**D**) Cauda sperm was collected from 8 weeks-old mice and visualized via SEM (n = 3 per genotype). Sperm with or without cytoplasmic droplets were counted. n = 250, per animal. (**E**) Increased size of cytoplasmic droplets in cKO/Y$^{Ngn3-Cre}$ sperm. Representative SEM images are shown. Upper panels: cKO/Y$^{Ngn3-Cre}$ and fl/Y control. Lower panel: cKO/Y$^{Sf1-Cre}$ and fl/Y control. Droplets are indicated by arrows. (**F**) Summary of cytoplasmic droplets in cKO/Y$^{Ngn3-Cre}$ sperm. Droplet surface size of SEM images was determined via ImageJ. n = 100, each. (**G**) Summary of IVF using sperm isolated from cKO/Y$^{Ngn3-Cre}$ and littermate control males. n = 262 and 324 presumed oocytes for fl/Y and cKO/Y sperm, respectively, seven animals, each. Values represent the mean ± s.e.m.

*Figure 7 continued on next page*

*Figure 7 continued*

The online version of this article includes the following figure supplement(s) for figure 7:

**Figure supplement 1.** Normal numbers of Sertoli cells in testes lacking *Rlim*.

droplets in cKO/Y$^{Ngn3-Cre}$ animals when compared with cKO/Y$^{Sf1-Cre}$ animals and their respective littermate controls (*Figure 7D*). Moreover, focusing on sperm with cytoplasmic droplets, the droplet sizes in cKO/Y$^{Ngn3-Cre}$ sperm were increased (*Figure 7E,F*), similar to those of cKO/Y$^{Sox2-Cre}$ sperm (*Figure 5C,D*). IVF experiments using swim-out sperm yielded in 76% cleavage stage and 58% blastocyst stage embryos for control, compared to 62% and 33% for cKO/Y$^{Ngn3-Cre}$ sperm, respectively (*Figure 7G*), confirming the functional deficiency of sperm. Thus, the phenotypes in males systemically lacking *Rlim* are largely recapitulated by cKO/Y$^{Ngn3-Cre}$ but not cKO/Y$^{Sf1-Cre}$ mice, demonstrating functions of *Rlim* in the spermatogenic cell lineage. Our combined results provide strong evidence that the upregulated expression of *Rlim* in round spermatids plays important functions for the reduction of cytoplasmic volume in sperm.

## Discussion

Our results reveal robust expression of *Rlim* in male reproductive organs, particularly in testis, where the expression pattern was highly dynamic both at the mRNA and protein levels (*Figures 1* and *2*). The appearance of a variant mRNA in males coincides with sexual maturation, suggesting that this form is predominantly expressed in differentiating round spermatids, which express high levels of RLIM protein (*Figure 2A*), and, because *Rlim* acts in the spermatogenic cell lineage (*Figure 7*), also coincides with the exertion of its in vivo function in males. Indeed, alternative mRNAs display different time of synthesis, mRNA stability and/or translational efficiency (*Tian and Manley, 2017*). At the protein level, we find RLIM expression in spermatogenic cells is post-meiotically upregulated from low levels in step three spermatids to high levels peaking in round spermatids at step 6–8 and then downregulated again in elongating step nine spermatids and thereafter (*Figure 2*). Thus, *Rlim* joins many X-linked genes that are reactivated after meiotic sex chromosome inactivation (MSCI) (*Ernst et al., 2019*) and it is tempting to speculate that this reactivation is connected with the occurrence of the alternative *Rlim* mRNA. Moreover, the downregulation of RLIM in elongating spermatocytes coincides with a change in transcriptional and chromatin dynamics at this stage (*Ernst et al., 2019*). Even though *Rlim* is X-linked, the finding that the RLIM protein is detected in most/all round spermatids, including those that presumably harbor a Y chromosome, is explained by the fact that cytoplasmic bridges exist between spermatids and that RLIM efficiently shuttles between nuclei in heterokaryon cells (*Jiao et al., 2013*).

Our RNA-seq analyses indicate that many genes are affected in testes lacking *Rlim* influencing various cellular functions including signaling and metabolism (*Figure 3*). As *Rlim* regulates transcriptional factors (*Bach et al., 1999*; *Ostendorff et al., 2002*; *Krämer et al., 2003*; *Güngör et al., 2007*; *Gontan et al., 2012*; *Johnsen et al., 2009*; *Her and Chung, 2009*; *Huang et al., 2011*; *Wang et al., 2019*), it is thus likely that many of the differentially expressed genes (*Figure 3D,E*) might be affected indirectly. Because the vast majority of cells in the testis reflect spermatogenic cells, where *Rlim* exerts its functions (*Figure 7*), many differentially expressed genes reflect those expressed in this cell lineage and therefore may collectively contribute to the observed *Rlim* KO phenotypes in testes. However, we cannot exclude the possibility that subtle and hence undetected changes in testicular cell types in mice lacking *Rlim* may contribute to the observed differences in gene expression as well as testes weight (*Figure 3*).

Sperm produced by males lacking *Rlim* is dysfunctional with decreased motility and increased cytoplasm and head abnormalities. Because excess cytoplasm affects sperm motility, morphology including head morphology as well as fertilization potential (*Cooper, 2011*; *Rengan et al., 2012*), and the cytoplasmic volume is regulated during spermiogenesis just after RLIM protein is highly detected in spermatids, it is likely that the increased cytoplasmic volume is responsible for much of the defects detected in sperm lacking *Rlim*. Moreover, as the midpiece cytoplasm is particularly important for sperm osmoregulation (*Cooper, 2011*; *Rengan et al., 2012*), an increased size of cytoplasmic droplets is predicted to render sperm more vulnerable to osmotic challenges, which may

ultimately lead to midpiece rupturing (*Figure 6*). Thus, while the finding of increased metabolite content in cKO sperm (*Figure 5A*) suggests defective cytoplasmic reduction during spermiation even though major defects were not observed (*Figure 7—figure supplement 1*), it is likely that the some of the increase in cytoplasmic size might have occurred in released sperm after spermiation. In this context, an increased sperm cytoplasm has been associated with higher activities of specific enzymes of the energy pathway including G6PDH (*Aitken et al., 1994*; *Yuan et al., 2013*), which diverts glucose metabolism away from glycolysis toward the pentose pathway. It is thus tempting to speculate that increased G6PDH activity might be partially responsible for the decreased Acetyl-CoA levels measured in sperm lacking *Rlim* (*Figure 5A*). Our combined data suggests that inefficient cytoplasmic reduction during spermiogenesis/spermiation renders *Rlim* KO sperm vulnerable to adequately adjust to the changing environment that occurs during transit through the epididymis (*Gervasi and Visconti, 2017*; *Sullivan and Saez, 2013*) leading to functional defects.

Concerning the cell type of *Rlim* action, RLIM protein is detected in spermatogenic cells specifically in round spermatids, in Sertoli cells and in epididymal epithelial cells. Our combined data reveals that lack of RLIM specifically in round spermatids is responsible for much of the observed sperm phenotype. This is demonstrated by targeting the *Rlim* cKO via Ngn3-Cre in the spermatogenic cell lineage, resulting in defective spermiogenesis, and similar sperm phenotypes when compared to the systemic *Rlim* cKO (*Figures 3* and *7*). In contrast, targeting the *Rlim* cKO via Sf1-Cre to Sertoli cells failed to induce a testis/sperm phenotype and providing strong evidence that *Rlim* in Sertoli cells is not involved in the regulation of spermiogenesis provided by this cell type. Thus, while we cannot exclude minor functions of *Rlim* in Sertoli cells and possibly also epididymal epithelial cells, our results provide strong evidence that *Rlim* in round spermatids is required for normal spermiogenesis. Therefore, *Rlim* adds to a very limited number of genes in the spermatogenic cell lineage that regulate the cytoplasmic reduction/droplet size (*Zheng et al., 2007*; *Mikolcevic et al., 2012*).

Considering early lethality of female mouse embryos with a maternally transmitted *Rlim* mutation (*Shin et al., 2010*), mathematical modeling of this exclusively female phenotype indicates that a deleterious mutation in the *Rlim* gene would shift selective evolutionary pressure entirely on females leading to a gender bias toward males in a mouse population over time (*Jiao et al., 2012*). Because gender biases in mouse populations represent an unfavorable strategy for reproduction (*Hamilton, 1967*), it is likely that the observed functions of *Rlim* during male reproduction will contribute to counter-act gender biases induced by its female function.

In summary, our data provide first evidence that in addition to crucial epigenetic functions in female embryogenesis and reproduction, the E3 ubiquitin ligase RLIM also occupies important roles during the reproduction of male mice. These results have major implications for epigenetic regulation and emphasize the importance of the UPS in male reproduction.

# Materials and methods

## Key resources table

| Reagent type (species) or resource | Designation | Source or reference | Identifiers | Additional information |
|---|---|---|---|---|
| Genetic reagent (*M. musculus*) | *Rlim^flox^* | PMID:20962847 | MGI:1342291 | |
| Genetic reagent (*M. musculus*) | Sox2-Cre | Jackson Laboratory | JAX #008454 | PMID:14516668 |
| Genetic reagent (*M. musculus*) | Neurog3-Cre (Ngn3-Cre) | PMID:15183725 | JAX #005667 | Dr. Andrew Leiter (UMMS) |
| Genetic reagent (*M. musculus*) | Sf1-Cre | Jackson Laboratory | JAX #012462 | PMID:16423694 |
| Antibody | Rabbit anti-Rlim (rabbit polyclonal) | PMID:11882901 | | IHC (1:250) WB (1:1000) |
| Antibody | Rat anti-GATA1 (rat monoclonal) | Santa Cruz | sc265 | IHC (1:100) |
| Antibody | Rabbit anti-GATA4 (rabbit polyclonal) | Abcam | ab84593 | IHC (1:500) |
| Antibody | Rabbit anti-phosphoPKA (rabbit monoclonal) | Cell Signaling clone 100G7E | 9624 | WB (1:10000) |
| Antibody | Mouse anti-PY (mouse monoclonal) | Millipore clone 4G10 | 05–321 | WB (1:10000) |

*Continued on next page*

*Continued*

| Reagent type (species) or resource | Designation | Source or reference | Identifiers | Additional information |
|---|---|---|---|---|
| Antibody | Mouse anti- β-actin (mouse monoclonal) | Santa Cruz | sc47778 | WB (1:250) |
| Antibody | Rabbit anti-cleaved caspase3 (rabbit monoclonal) | Cell Signaling | ab9664 | IHC (1:300) |

## RNA-seq and data analyses

RNA-seq on RNA isolated from testes of fl/Y and cKO/Y males including library construction and sequencing on a NextSeq 500 was carried out essentially as described (*Wang et al., 2016*; *Wang et al., 2017*). Reads (paired end 35 bp) were aligned to the mouse genome (mm10) using TopHat (version 2.0.12) (*Trapnell et al., 2009*), with default setting except set parameter read-mismatches was set to 2, followed by running HTSeq (version 0.6.1p1) (*Anders et al., 2015*), Bioconductor packages edgeR (version 3.10.0) (*Robinson et al., 2010*; *Robinson and Smyth, 2007*) and ChIPpeakAnno (version 3.2.0) (*Zhu, 2013*; *Zhu et al., 2010*) for transcriptome quantification, differential gene expression analysis, and annotation. For edgeR, we followed the workflow as described in *Anders et al., 2013*.

## De-lipidation of epididymis

De-lipidation of epididymis was performed based on previously described protocols (*Sylwestrak et al., 2016*; *Tomer et al., 2014*). Briefly, isolated organs were fixed with 4% paraformaldehyde (PFA) in PBS for 32 hr at room temperature (RT), then rinsed with PBS for three times of at least 2 hr. Tissues were kept at 4°C in PBS with 0.02% sodium azide until the time of tissue processing. In order to visualize tubules that form the inner layers of the epididymis, a de-lipidation step was performed. De-lipidation was done passively by incubating the organ with 4% SDS/PBS at RT in an orbital shaker for 2 weeks. The 4% SDS/PBS solution was changed every other day. At the end of the second week, the organ was rinsed with PBS for three times of least 4 hr and finally placed on a refractive index matching solution (RIMS: 0.17M iodixanol; 0.4M diatrizoic acid, 1M n-methyl-d-glucamine, 0.01% sodium azide), 24 hr prior to imaging.

## Collection of sperm

Epididymal Cauda sperm collected via swim-out and testicular sperm which were analyzed in this study was collected from 8 weeks-old fl/Y or cKO/Y mice. Briefly, Cauda epididymides were dissected and placed in 1 ml of modified Krebs-Ringer medium (m-TYH; 100 mM NaCl, 4.7 mM KCl, 1.2 mM $KH_2PO_4$, 1.2 mM $MgSO_4$, 5.5 mM Glucose, 0.8 mM Pyruvic Acid, 1.7 mM Calcium Chloride, 20 mM HEPES). Sperm was allowed to swim-out for 10 min at 37°C and then the epididymides were removed. Concentration of all sperm was calculated using a Neubauer hemocytometer. For mature testicular spermatozoa isolation, testes from one 8-week-old mouse were minced in a 35 mm Petri dish containing 1 ml 150 mM NaCl. Finely minced tissue slurry was then transferred to a 15 ml conical tube and set aside for 3–5 min to allow tissue pieces to settle down. Next, the cell suspension was loaded onto 10.5 ml of 52% isotonic percoll (Sigma). The tubes were then centrifuged at 15000 x g for 10 min at 10°C. The pellet was resuspended in 10 ml of 150 mM NaCl and spun at 900 x g at 4°C for 10 min followed by three washes with 150 mM NaCl at 4000 x g at 4°C for 5 min.

## Sperm analyses

Concerning the assessment of sperm morphology, after swim-out, sperm suspensions (50 µl) were fixed with paraformaldehyde 4% (w/v; EMS, Hatfield, MA) in phosphate buffered saline (PBS) for 10 min at room temperature. Then samples were centrifuged at 800 x g for 5 min, washed with PBS, and let air-dry on poly-L-lysine-coated glass slides. Samples were mounted using VectaShield mounting media (H-1000, Vector Laboratories, Burlingame, CA) and differential interference contrast (DIC) images were taken using a 60X objective (Nikon, PlanApo, NA 1.49) in an inverted microscope (Nikon Eclipse TE300). Images were analyzed using the freeware ImageJ v1.52e (http://imagej.nih.gov/ij/download.html), and sperm morphology was classified into the following categories: normal, coiled midpiece, head malformation, flipped head, and residual cytoplasmic droplet. Results are shown as percentages, at least 200 sperm per sample were counted in single-blinded experiments. Concerning the analysis of sperm acrosomal status, sperm from the swim-out suspension were

loaded into glass slides and let air-dry for 15 min. After that samples were fixed, and the acrosomes were stained as described below in the sperm acrosome reaction section. Fluorescence and phase contrast images were taken in a Nikon Eclipse TE300 fluorescence microscope using a 40x objective (Nikon, Phase 2 DL, NA 0.55). Images were analyzed using the freeware ImageJ v1.52e, and at least 200 sperm per sample were counted in single-blinded experiments. Results are shown as percentage of acrosome intact sperm.

Concerning sperm capacitation and analysis of swim-out cauda sperm motility, sperm were incubated at 37°C for 60 min in m-TYH (Non-Cap) or in m-TYH supplemented with 15 mM NaHCO$_3$ and 5 mg/ml BSA (Cap). Sperm motility was evaluated in the swim-out (T = 0) and after 60 min of incubation in capacitating conditions (T = 60). Briefly, sperm suspensions (30 µl) were loaded into prewarmed chamber slides (Leja slides, Spectrum Technologies, Healdsburg, CA) and placed on a warmed microscope stage at 37°C. Sperm motility was examined using the CEROS computer-assisted semen analysis (CASA) system (Hamilton Thorne Research, Beverly, MA). Acquisition parameters were set as follows: frames acquired: 90; frame rate: 60 Hz; minimum cell size: four pixels; static head size: 0.13–2.43; static head intensity: 0.10–1.52; and static head elongation: 5–100. At least five microscopy fields corresponding to a minimum of 200 sperm were analyzed in each experiment. Data were analyzed using the CASAnova software (*Goodson et al., 2011*).

Concerning SDS-PAGE and western blotting, swim-out cauda sperm samples were centrifuged at 12,000 x g for 2 min, washed in 1 ml of PBS, and then centrifuged at 12,100 x g for 3 min. Sperm proteins were extracted by resuspending the remaining pellets in *Laemmli, 1970* sample buffer, boiled for 5 min and centrifuged once more at 12,100 x g for 5 min. Protein extracts (supernatant) were then supplemented with β-mercaptoethanol 5% (v/v), and boiled again for 4 min. Protein extracts equivalent to 2.5 × 10$^5$ sperm/lane were subjected to SDS–PAGE, and electro-transferred to PVDF membranes (Bio-Rad, Waltham, MA). PVDF membranes were blocked with 5% (w/v) fat-free milk in tris buffered saline containing 0.1% (v/v) Tween 20 (T-TBS) and immunoblotted with anti-pPKAs antibody (clone 100G7E, 1:10,000) overnight at 4°C to detect phosphorylated PKA substrates. Then, membranes were incubated with HRP-conjugated anti-rabbit secondary antibody diluted in T-TBS (1:10,000) for 60 min at room temperature. Detection was done with an enhanced chemiluminescence ECL plus kit (GE Healthcare) as per manufacturer instructions. After developing of pPKAs, membranes were stripped at 55°C for 20 min in 2% (w/v) SDS, 0.74% (v/v) β-mercaptoethanol, 62.5 mM Tris (pH 6.5), blocked with fish gelatin 20% (v/v; Sigma cat # G7765, St. Louis, MO) in T-TBS for 60 min at room temperature and re-blotted with anti-PY antibody (clone 4G10, 1:10,000) to detect proteins phosphorylated in tyrosine residues. Membranes were then incubated with HRP-conjugated anti-mouse secondary antibody diluted in T-TBS (1:10,000) for 60 min at room temperature. Detection was done with an enhanced chemiluminescence ECL plus kit (GE Healthcare) as per manufacturer instructions.

Concerning swim-out cauda sperm acrosome reaction, after 60 min of capacitation in m-TYH Cap medium, sperm samples were incubated with progesterone (10 µM) or with DMSO (vehicle) in m-TYH Cap at 37°C for 30 min. Then, sperm were loaded into glass slides and let air-dry for 15 min. Sperm were fixed by incubation with paraformaldehyde 4% (w/v) in PBS at room temperature for 15 min, washed three times (5 min each) with PBS and permeabilized with 0.1% (v/v) Triton X-100 for 3 min. After permeabilization, samples were washed three times with PBS, and then incubated with Alexa Fluor 488-conjugated lectin peanut agglutinin (PNA) in PBS at room temperature for 30 min. Before mounting with Vectashield (Vector Laboratories, Burlingame, CA), samples were washed three times with PBS for 5 min each time. Epifluorescence images were taken in a Nikon Eclipse TE300 fluorescence microscope using a 40x objective (Nikon). Phase contrast images were taken in parallel. Images were analyzed using the freeware ImageJ v1.52e, and at least 200 sperm per sample were counted in single-blinded experiments. Results are shown as percentage of acrosome reacted sperm.

## In vitro fertilization (IVF)

IVF experiments were performed essentially as described (*Sharma et al., 2016*) with some modifications. Briefly, female C57BL/6J mice (age 6–8 weeks) were superovulated by intraperitoneal injection of pregnant mare's serum gonadotropin (PMSG; 5 U; Calbiochem) followed by human chorionic gonadotropin (hCG; 5 U; Sigma-Aldrich) 48 hr later (*Yamashita et al., 2008*). Metaphase II-arrested oocytes tightly packed with cumulus cells were collected from the oviductal ampulla 14 hr after hCG

injection and placed in a 100 µl drop of human tubal fluid (HTF; Millipore) medium covered with mineral oil. Fresh cauda epididymal sperm of fl/Y and cKO/Y littermates (>6 animals per genotype both for Sox2-Cre and Ngn3-Cre, aged 2–4 mo) were swum-up in 1 ml HTF medium for 10 min and capacitated by incubation for another 30 min at 37°C under 5% $CO_2$. An aliquot ($1.0 \times 10^5$ cells) of the capacitated sperm suspension was added to 100 µl drop of HTF medium containing the oocytes. After incubation at 37°C under 5% $CO_2$ for 4 hr, the presumed zygotes were washed with KSOM medium to remove cumulus cells, sperm and debris, and then incubated in a 50 µl drop of KSOM medium. In vitro fertilized embryos were analyzed at cleavage (24 hr) and blastocyst stages (96 hr).

## Metabolomic profiling

Polar metabolite measurements were performed by the Metabolite Profiling Core Facility at the Whitehead Institute for Biomedical Research (Cambridge, MA). For LC-MS analyses, 45 Mio epididymal cauda swim-out sperm were collected each from 5 cKO/Y$^{Sox2-Cre}$ and three fl/y animals in extraction mix containing 80% methanol and isotopically labeled amino acids. Peak area ratios of each metabolite were determined and $log_2$ ratio relative to the mean of control (fl/Y) sperm determined.

## Electron microscopy

SEM was performed on caudal epididymis sperm and sperm isolated from testes. Caudal sperm was let to dissociate in HTF medium and then fixed in 2.5% glutaraldehyde, 2% paraformaldehyde in 0.1M Na Cacodylate buffer. The fixed sperm was then placed on poly Lysine covered cover slips and left to adhere for 10 min. The samples were rinsed three times in the same fixation buffer, post fixed in aqueous 1% (w/v) OsO4 for 1 hr at RT, dehydrated through a graded ethanol series to ethanol 100% (x3), and then they were critically point dried. The cover slips were then mounted on double sided carbon tape onto aluminum SEM stubs and grounded with colloidal silver paint, sputter coated with 12 nm of gold-palladium and were imaged using secondary electron (SEI) mode with a FEI Quanta 200 MKII FEG SEM.

TEM was performed on ultrathin sections on Caput epididymides and testes. Tissues were dissected and immediately immersed in 2.5% glutaraldehyde in 0.1 M Na Cacodylate buffer, pH 7.2 for 60 min at RT. The samples were rinsed three times in the same fixation buffer and post-fixed with 1% osmium tetroxide for 1 hr at room temperature. Samples were then washed three times with ddH$_2$O for 10 min, and in block stained with a 1% Uranyl Acetate aqueous solution (w/v) at 6°C overnight. After three rinses in ddH2O the samples were dehydrated through a graded ethanol series of 20% increments, before two changes in 100% ethanol. Samples were then infiltrated first with two changes of 100% Propylene Oxide and then with a 50%/50% propylene oxide / SPI-Pon 812 resin mixture. The following day 5 changes of fresh 100% SPI-Pon 812 resin were done before the samples were polymerized at 68°C in flat embedding molds. The samples were then reoriented, and thin sections (approximately 70 nm) were placed on copper support grids and contrasted with Lead citrate and Uranyl acetate. Sections were examined using the a CM10 TEM with 100Kv accelerating voltage, and images were captured using a Gatan TEM CCD camera.

## Acknowledgements

We are grateful to M Krykbaeva, L Huang, A Schlueter, G Pazour, J Shin and E Torres for advice and/or experimental help. We thank A Leiter for providing Ngn3-Cre mice and the Metabolite Profiling Core Facility at the Whitehead Institute for running metabolomics samples and for data analysis. IB is a member of the University of Massachusetts DERC (DK32520). This work was supported from NIH grants R01GM128168 to IB, R01HD080224 and DP1ES025458 to OJR, and R01HD38082 to PEV.

## Additional information

### Funding

| Funder | Grant reference number | Author |
| --- | --- | --- |
| National Institutes of Health | GM128168 | Ingolf Bach |

| National Institutes of Health | HD080224 | Oliver J Rando |
| National Institutes of Health | HD038082 | Pablo E Visconti |
| National Institutes of Health | DP1ES025458 | Oliver J Rando |
| University of Massachusetts | DK32520 | Ingolf Bach |

The funders had no role in study design, data collection and interpretation, or the decision to submit the work for publication.

## Author contributions
Feng Wang, Conceptualization, Formal analysis, Investigation, Methodology, Writing - original draft; Maria Gracia Gervasi, Formal analysis, Investigation, Methodology, Writing - original draft; Ana Bošković, Vera D Rinaldi, Investigation, Methodology; Fengyun Sun, Jun Yu, Mary C Wallingford, Darya A Tourzani, Lara Strittmatter, Investigation; Jesse Mager, Lihua Julie Zhu, Oliver J Rando, Pablo E Visconti, Supervision; Ingolf Bach, Conceptualization, Formal analysis, Supervision, Funding acquisition, Investigation, Writing - original draft, Writing - review and editing

## Author ORCIDs
Maria Gracia Gervasi (iD) https://orcid.org/0000-0002-5468-2700
Vera D Rinaldi (iD) http://orcid.org/0000-0002-0051-1754
Oliver J Rando (iD) http://orcid.org/0000-0003-1516-9397
Ingolf Bach (iD) https://orcid.org/0000-0003-4505-8946

## Ethics
Animal experimentation: All mice were housed in the animal facility of UMMS and utilized according to NIH guidelines and those established by the UMMS Institute of Animal Care and Usage Committee (IACUC; protocol #201900344).

## Decision letter and Author response
Decision letter https://doi.org/10.7554/eLife.63556.sa1
Author response https://doi.org/10.7554/eLife.63556.sa2

# Additional files

## Supplementary files
• Transparent reporting form

## Data availability
RNAseq data have been deposited in GEO under accession code GSE114593.

The following dataset was generated:

| Author(s) | Year | Dataset title | Dataset URL | Database and Identifier |
|---|---|---|---|---|
| Wang F, Bach I | 2020 | Analysis of functions of Rlim during reproduction in male mice | https://www.ncbi.nlm.nih.gov/geo/query/acc.cgi?acc=GSE114593 | NCBI Gene Expression Omnibus, GSE114593 |

The following previously published dataset was used:

| Author(s) | Year | Dataset title | Dataset URL | Database and Identifier |
|---|---|---|---|---|
| Margolin G, Khil PP, Bellani MA, Camerini-Otero RD | 2014 | RNA-Seq and RNA Polymerase II ChIP-Seq of mouse spermatogenesis | https://www.ncbi.nlm.nih.gov/geo/query/acc.cgi?acc=GSE44346 | NCBI Gene Expression Omnibus, GSE44346 |

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
