## [Decision Letter]

**Acceptance summary:**

The new insights into *Rlim* and the unexpected role that it plays in male spermatogenesis will be of interest to readers of eLife.

**Decision letter after peer review:**

[Editors’ note: the authors submitted for reconsideration following the decision after peer review. What follows is the decision letter after the first round of review.]

Thank you for submitting your work entitled "Deficient spermatogenesis and sperm maturation in mice lacking *Rlim*" for consideration by *eLife*. Your article has been reviewed by three peer reviewers, and the evaluation has been overseen by a Reviewing Editor and a Senior Editor. The reviewers have opted to remain anonymous. Our decision has been reached after consultation between the reviewers. Based on these discussions and the individual reviews below, we cannot proceed with the current form of the manuscript.

Your manuscript revisits the effect of *RLIM* deficiency on male fertility. Previously, your work showed that *Rlim* deletion has no obvious effect on male fertility (Shin et al., 2010), but the current study shows expression of a testis-specific *Rlim* transcript variant with a shorter 3' UTR and more detailed study shows that KO males have smaller testes, epididymides, fewer sperm, more frequent morphological abnormalities, lower motility, and aberrant gene expression. Furthermore, embryos generated from in vitro fertilisation using *Rlim* KO sperm progress to blasts at a lower rate.

These are interesting observations, but all reviewers find the study to be on the preliminary side and all agree that a characterization of mechanism would be required for publication in *eLife*. The reviewers pointed out a number of additional technical issues that would need to be addressed. We recognize that you might be able to perform additional work to shed light on mechanism and address technical concerns. We would ordinarily only proceed to a revised manuscript if the amount of work required for revision can be managed in 1-2 months. In this case, we feel that much more than 2 months would be needed. Although we reject the current version of the paper, we invite you to resubmit the manuscript should you be able to perform additional experiments to elucidate how *Rlim* defects can lead to defective spermatogenesis. We do not know what the reviewers specifically have in mind, but we imagine that something like an RNA-seq analysis that identifies differentially expressed genes involved in KO versus control sperm at the level of metabolomics and/or lipidomics. Alternatively, a mass spec analysis for differentially ubiquinated proteins or differences in protein profiles might be helpful.

Reviewer #1:

In previous work, the Bach lab showed that *Rlim* deletion has no obvious effect on male fertility (Shin et al., 2010). In this follow-up manuscript, they show that *RLIM* protein is expressed in pachytene spermatocytes, Sertoli cells (testis) and epithelial cells (epididymis). Northern blot and RNA-seq analyses reveal expression of a testis-specific *Rlim* transcript variant with a shorter 3' UTR. Loss of function of *Rlim* using *Sox2*-Cre-induced whole embryonic knockout (KO) shows that *Rlim* KO males have smaller testes, with dysregulated gene expression. *Rlim* KO males also have smaller epididymides, which contain fewer sperm, more frequent morphological abnormalities and lower motility. Accordingly, embryos generated from in vitro fertilisation using *Rlim* KO sperm develop to blastocysts at a lower rate compared to control.

My comments are as follows:

1) In its current form the study doesn't provide mechanistic insight. The analysis is incremental and would not contribute to the significant advance of reproductive biology. Therefore, whether the findings are of the highest scientific importance which *eLife* aims to publish is unclear.

2) Figure 2A: western blot data of *RLIM* should be added. If the short variant has a full-length ORF, protein size would be the same in testis and other tissues.

3) Figure 3A, B: please indicate age of the mice used for analysis. What does "testis 1/2" mean in B? If just single testis of each mouse was weighed, the data might be biased and mean weight of both testes should have been used. Is apoptosis in *Rlim* KO testis observed as it could explain smaller testis size?

4) Figure 3C-E: it is unclear whether gene dysregulation in the KO occurs in germ cells or other cell types. The authors could sort different stages of spermatogenic cells in testis (example shown in Bastos et al., Cytometry Part A, 2005) and perform RNA-seq analysis.

5) Figure 3—figure supplement 1: the macro-H2A1 data is not convincing, and looks like background staining. I would suggest performing immunofluorescence staining of meiocyte nuclear spreads. Also, marker protein expression doesn't reveal whether XY silencing is affected. The authors' RNA-seq data could be used to ask whether sex-genes are de-repressed or not. Again, cell-type specific analysis suggested in comment 3 is important as the authors can focus the analysis on pachytene cell population, in which sex chromosome silencing occurs.

6) Figure 4-6: the relationship between *RLIM* expression and the spermiogenesis phenotype is not clear. This mechanistic deficiency could be addressed using Cre lines that deplete *Rlim* in specific cell types (e.g. pachytene, Sertoli cells).

Reviewer #2:

This study reported a role of *Rlim* in male reproduction. *RLIM* is a RING finger ubiquitin E3 ligase. Previous studies have shown that *Rlim* knockout females are embryonic lethal due to a failure in the maintenance of X-inactivation in extraembryonic tissues but KO males are fertile. In the current study, *Rlim* expression was examined in both testis and epididymis at both protein and RNA levels. It was concluded that *RLIM* protein level peaks in pachytene spermatocytes. The *Rlim* KO males show reduced sperm count and decreased testis weight. The KO sperm display reduced motility and reduced fertilization rate in vitro. RNA-seq on adult KO and control testes revealed altered transcriptome. While sperm and fertilization defects are well characterized and interesting, the overall study is very preliminary. It is unclear how inactivation of *Rlim* causes such sperm defects. No attempt was made to identify the *RLIM* target proteins in testis. The fact that the *Rlim* KO males are still fertile diminishes its significance. There is a major problem with data interpretation. Overall, I don't feel that this preliminary study represents a significant contribution.

1) Expression of *RLIM* protein in testis: The expression data interpretation (Figure 1) is not correct. It was concluded that *RLIM* expression peaks in pachytene spermatocytes at stages VI/VII (Figure 1B). *RLIM*-positive cells are also PNA-positive (acrosomal caps) and thus are round spermatids instead of pachytene spermatocytes. *RLIM*-positive round spermatids are obvious in the Figure 1A-1 tubule. Based on the data presented, the correct interpretation is that *Rlim* is expressed in round spermatids and Sertoli cells in the testis. While this mistake can be easily corrected, it has impacted the entire manuscript in a wrong way, which needs to be re-interpreted and re-written.

2) Figure 2 shows the generation of a short testis-specific transcript due to alternative polyadenylation. This is quite common in testis and is not really meaningful.

3) Impact on MSCI. First, *Rlim* is not expressed in pachytene spermatocytes (see point #1), thus discussion of MSCI is not relevant. Second, gammaH2AX and macroH2A1 are very abundant, their presence may not necessarily indicate lack of effect on MSCI. It is better to look at the expression of sex-linked (X-linked and Y-linked) genes from their RNA-seq data. If MSCI is not affected, sex-linked genes should not be preferentially affected.

4) The RNA-seq was performed on adult whole testis instead of enriched germ cell types. Given that the cKO testis is smaller and the sperm count is lower, differential expression of genes (DEGs) identified could be due to the reduced proportion of germ cells in the KO testis. This needs to be taken into consideration.

Reviewer #3:

Wang et al. identified the localization of *RLIM* in mouse testes and epididymis. They also made *Rlim*cKO/Y mice and found decreased sperm production with high rates of morphological abnormalities as well as decrease in motility and fertilization rates in vitro. This study uncovered functions of *RLIM* in male reproduction, which may indicate potential evolutionary pressure for this gene. However, the study did not well illustrate the mechanism of the gene's functions in spermatogenesis, and some results they showed are not well analyzed. Thus, I suggest this paper to be rejected.

1) Though the authors showed abnormal sperm morphology and decreased sperm motility in *Rlim*cKO/Y mice, they did not identify the exact stage for the appearance of the phenotype, e.g. meiosis, post meiosis or during fertilization. Thus, they did not know the functional stage of this gene, which made it impossible for mechanism studies.

2) The authors suggested that *RLIM* has "high and dynamic expression in specific cell types including Sertoli cells as well as differentiating spermatocytes at the pachytene stage". However, according to the staining pattern of *RLIM* in Figure 1B, *RLIM* is exclusively localized in germ cells expressing PNA, which is only expressed in post meiotic cells. Thus, *RLIM* is not expressed in "differentiating spermatocytes at the pachytene stage".

3) Most of the major conclusions raised in this study are supported by a single experiment, which are not very convincing.

[Editors’ note: further revisions were suggested prior to acceptance, as described below.]

Thank you for submitting your article "Deficient spermiogenesis in mice lacking *Rlim*" for consideration by *eLife*. Your article has been reviewed by three peer reviewers, and the evaluation has been overseen by Patricia Wittkopp as the Senior and Reviewing Editor. The following individual involved in review of your submission has agreed to reveal their identity: Julie Cocquet (Reviewer #3).

The reviewers have discussed the reviews with one another and the Reviewing Editor has drafted this decision to help you prepare a revised submission.

Summary:

This paper examines the role of *Rlim* E3 ubiquitin ligase during mouse spermatogenesis. The authors show that spermatozoa from *Rlim* knockout (KO) males are malformed, less motile and present with a defect in cytoplasmic reduction leading to a small decrease in their fertilizing abilities. They looked at *Rlim* testicular expression and identified a smaller transcript isoform which is highly expressed in postmeiotic cells. Expression in epididymis epithelium was also observed but its role remains unclear. Finally by producing a male germ cell specific KO, the authors conclude that *Rlim* postmeiotic isoform is at the basis of the observed sperm defects in the KO.

This is a resubmission of work rejected previously because it failed to elucidate how *Rlim* defects can lead to defective spermatogenesis. While all three current reviewers (two of which were not involved in the initial evaluation) applauded the addition of new data (i.e., additional data of metabolite and detailed morphology analyses of *Sox2*-Cre *Rlim* KO sperm, as well as conditional *Rlim* KO analyses using Cre lines specific to testicular germ cells or Sertoli cells), they also felt that the issues raised in the first round of revision (e.g., the molecular function of *Rlim* in spermatids) had not been fully addressed. They were optimistic, however, that they could be addressed with another round of revision addressing the points below.

Essential revisions:

1) Reviewer 1, point 1) The authors showed that germline *Rlim* KO phenocopies the *Sox2*-Cre *Rlim* KO phenotype in sperm (Figure 7). In this experiment, quantification of *RLIM* protein decrease in the target cell types (germ cells in Ngn3-Cre KO and Sertoli cells in Sf1-Cre KO) should be shown. Also, is the IVF phenotype (Figure 4 F-G) also observed in the Ngn3-Cre *Rlim* KO? This will provide more important and direct evidence than the morphology analyses in Figure 7 to show sperm defects in the Ngn3-Cre *Rlim* KO.

2) Reviewer 1, point 2) Figure 2A: The data requested was *RLIM* western blot using wildtype tissues to compare the *RLIM* protein size of the testis-specific short variant and the long variant expressed in other tissues. Because ORF is shared in these transcripts, their protein size would be same. This data should be added in the revision.

3) Reviewer 1, point 9) It is stated that "the Cauda region of many cKO/YSox2-Cre animals contained thinner tubules when compared to control males". The requested data was quantification of this thinner tubule phenotype. This data should be added in the revision.

Reviewer 1, point 10) Figure 4—figure supplement 1F suggests that *Rlim* deletion in sperm affects embryonic development after fertilisation. This point should be discussed more.

4) Throughout the Introduction including on the first line there is in appropriate reference to adult spermatogenesis involving primordial germ cells. These are not present in adult testes – should be undifferentiated spermatogonia or spermatogonial stem cells instead e.g. the progression of PGCs to mature spermatozoa should be progression of spermatogonia; stem cells to mature spermatozoa

5) First paragraph of Introduction – midpiece is a section of the sperm tail so should read “formation of the sperm head and tail” or to be more precise “the formation of the sperm head, HTCA and tail” not “formation of the sperm head, midpiece and tail”

6) First paragraph of Introduction – “the number of genes involved remain limited.” Should be “the number of genes identified to be involved remains limited.”

7) When describing which spermatid population *RLIM* is expressed in should use spermatid step not just seminiferous tubule stage (as some tubules have two types of spermatids in them so it is better practice) e.g. the spermatids that it is expressing in stage VI/VII – are step 6-7 spermatids. Also, the meaning would be clearer especially to non-specialists if you change the following sentence:

“*RLIM* levels are dramatically upregulated specifically in round spermatids that have undergone meiosis at stages VI / VII, an early timepoint in spermiogenesis (O'Donnell, 2014; Qian et al., 2014).”

to

“*RLIM* levels are dramatically upregulated specifically in post-meiotic step 6-7 round spermatids (stages VI / VII), an early timepoint in spermiogenesis (O'Donnell, 2014; Qian et al., 2014).”

As previous wording could be read to mean that round spermatids undergo meiosis at stage VI / VII which is of course incorrect. Also step 6-7 isn't super early in spermiogenesis it's about 1/3-1/2 to half the way through, what is notable is it is immediately before spermatids begin to elongate in step 8. While you make reference to the fact that its expression is no longer detected in spermatozoa that are released during spermiation – it would also be good to make reference in the results text to the fact that its expression in other spermatid steps is low e.g. once they begin to elongate it goes down and beforehand in step 3 it is low. More precise definition of the all the spermatids steps it is upregulated in would be beneficial

8) Figure 1B bottom panel is incorrectly staged. Based on acrosome morphology that is likely stage XI not IX, as the acrosome morphology shows that the apical hook/falciform shape has been acquired

9) More care needs to be taken with basic fertility phenotyping and characterisation of spermatogenesis. For example:

a) The use of swim out from the epididymis is not a quantitative method of sperm numbers as it is dependent on the cuts made as to how many sperm vacate, in addition non-motile sperm will be less likely to vacate the epididymis. Accurate sperm counts require homogenisation of the epididymis, particularly given the KO mice have lower motility the intrinsic bias of the sperm swim out towards motile sperm may results in KO sperm counts appearing to be lower than they really are

b) Basic PAS analysis of testis histology of KO testes should be conducted, with particular attention to stage 8 and 9 to see if there are defects in spermiation i.e. are all sperm properly released or do you see spermatid retention at stage 9, do residual bodies form normally, normal size etc (this would help define whether the origin of the cytoplasmic droplet is due to defects during spermiation or if it is related to defects in the normal changes that occur to it during transit through the epididymis). In addition of the steps within spermatids undergo head shaping to identify the origin of these defects.

c) Despite what the previous reviewer 3 said, testis and epididymis weight should not be expressed relative to body weight

d) The double axoneme phenotype is not commonly seen in KO models and the origins of this might reveal important insight about sperm tail formation. Have the authors characterized the head to tail coupling apparatus of these sperm and investigated if this is due to supernumerary basal bodies?

10) The authors need to include data showing validation of their KO models at an mRNA and protein level

11) How do the authors think the gene transcription changes in the *RLIM* KO mice relates to the phenotypes observed? Some interpretation of this is needed in the Discussion.

12) Result section *Rlim* expression in testis is highly regulated.

a) It is not clearly presented when exactly *RLIM* is highly expressed in postmeiotic cells. Is it restricted to stages VI to VIII? Since at stage IX on Figure 2B the signal is no longer strong (but see my comment below regarding staging). On Figure 2A, it seems as if many tubules display strong round spermatid signal. A precise description of *RLIM* protein dynamic at various stages of spermatogenesis is needed. Recapitulating IF/IHC observations using a scheme showing the different stages of spermatogenesis would be useful.

b) Figure 2

A – IHC panel:

Higher magnification insets from negative CTL would be useful to confirm that the signal observed in Sertoli cells is specific (i.e. not seen in neg control)

B - To me (but the figure is small) the image labelled "stage IX" is not a stage IX. I do not recognize the lectin PNA signal to be a stage IX: At this stage round spermatids have just started to elongate and condensed step 16 spermatids have already undergone spermiation (so are no longer present).

NB. Higher magnification of the DAPI images (in the merged figure for instance) would be helpful.

c) It seems odd to present *Rlim* expression in epididymal cells in the part entitled "*Rlim* expression in testis is highly regulated"

13) Result section Diminished production and functionality of *Rlim* KO sperm

a) Overall I find this part a bit confusing. At first, the testicular impairment seems to be minimized because of the observation that *Rlim* is expressed in epididymis epithelium. Yet it clearly appears from the last result section that sperm defects originate from a testicular (germ cell) impairment. Maybe this result section could be slightly re-organized?

specific comments/questions:

b) The observation of a small but significant reduction in testis weight indicates a (mild) spermatogenesis defect; so do testicular sperm morphological abnormalities.

Can the authors further investigate testicular cell composition – to see if one particular (likely postmeiotic) cell stage is affected? In the Discussion, it is said that increased cell death and chromatin packaging defects were investigated – even if the results are negative it would be worth presenting (or evoking) them in this section.

Since spermatogenesis is mildly impaired, RNAseq on whole testis should be interpreted with caution, as it could reflect a mild change in cell population. This emphasizes the need for a more detailed presentation of the testicular defects (or absence of visible defects).

c) Sentence: "…revealed that the Cauda region of many cKO/YSox2-Cre animals contained thinner tubules when compared to control males (Figure 3—figure supplement 1E), consistent with decreased Caudal sperm (Figure 3C)."

This sentence suggests that decreased caudal sperm is due to caudal region malformation. But to conclude, one needs to show quantification of testicular spermatozoa (or caput spermatozoa). Figure 6A data could be presented at that point.

But then I am confused: what is the authors' conclusion? is the decreased sperm count a consequence of testicular or epididymis defects? Since epididymis defects appear to be ruled out by the analysis of Ngn3-Cre KO sperm, it would increase clarity if epididymis analyses were presented separately.

14) Discussion

a) The first part about *Rlim* short isoform could be reduced. While it is an interesting fact, I feel it does not require so much emphasis since testis specific isoforms are quite common.

b) The fact that *Rlim* is X-encoded and expressed in spermatids does not mean it escapes MSCI (as MSCI occurs in pachytene primary spermatocytes). Instead, it means that *Rlim* short isoform is one of the many X-linked genes that are expressed after meiosis, when the X is re-activated (see for instance Ernst et al., 2019)

c) Considering the fact that the KO impact on male reproductive abilities (at least for the lab mouse) is minor, I find the part about the evolutionary consequences of a spermatogenesis role for *Rlim* a bit far-fetched. The paternal effect of *Rlim* in milk-producing alveolar cells would likely be more predominant (Jiao et al., 2012). I would recommend to put less emphasis on that aspect, especially in the Abstract.

---

## [Author Response]

[Editors’ note: the authors resubmitted a revised version of the paper for consideration. What follows is the authors’ response to the first round of review.]

Reviewer #1:In previous work, the Bach lab showed that Rlim deletion has no obvious effect on male fertility (Shin et al., 2010). In this follow-up manuscript, they show that RLIM protein is expressed in pachytene spermatocytes, Sertoli cells (testis) and epithelial cells (epididymis). Northern blot and RNA-seq analyses reveal expression of a testis-specific Rlim transcript variant with a shorter 3' UTR. Loss of function of Rlim using Sox2-Cre-induced whole embryonic knockout (KO) shows that Rlim KO males have smaller testes, with dysregulated gene expression. Rlim KO males also have smaller epididymides, which contain fewer sperm, more frequent morphological abnormalities and lower motility. Accordingly, embryos generated from in vitro fertilisation using Rlim KO sperm develop to blastocysts at a lower rate compared to control.My comments are as follows:1) In its current form the study doesn't provide mechanistic insight. The analysis is incremental and would not contribute to the significant advance of reproductive biology. Therefore, whether the findings are of the highest scientific importance which eLife aims to publish is unclear.

We believe that the revised manuscript provides much more insight into the mechanisms underlying the reproduction phenotype in male mice lacking *Rlim*. Because *Rlim* KO sperm displays decreased motility we have investigated the energy status of sperm polar metabolites and surprisingly found increased metabolite content (Figure 5). Detailed EM analyses reveals that cytoplasmic reduction is inhibited in *Rlim* KO sperm (Figures 5, 6) and using specific Cre drivers to target the *Rlim* cKO to Sertoli cells or the spermatogenic cell lineage we continue to show that *Rlim* in round spermatids is required for normal spermiogenesis (Figure 7). Using germline KO we show that the testis phenotype occurs independently of the expression of Cre-recombinase (Figure 3—figure supplement 1B, C). Combined these results identify *Rlim* as a novel and important regulator of cytoplasmic reduction in sperm, a process that is not well studied despite its major impact on male reproduction, thereby illuminating the evolution of the *Rlim* gene.

2) Figure 2A: western blot data of RLIM should be added. If the short variant has a full-length ORF, protein size would be the same in testis and other tissues.

Unfortunately, in our experience Western blot data do not allow distinguishing small differences in MW of *Rlim* protein, e.g. *Rlim* deletion mutants in which the NES or NLS is deleted (lacking 13 or 20 amino acids) migrate at a very similar position than the WT protein (see also Jiao et al., 2013; Figure 6B).

3) Figure 3A, B: please indicate age of the mice used for analysis. What does "testis 1/2" mean in B? If just single testis of each mouse was weighed, the data might be biased and mean weight of both testes should have been used. Is apoptosis in Rlim KO testis observed as it could explain smaller testis size?

All analyses on sperm have been carried out from sperm originating from 8 weeks old males. Even though we have included both testes halves from all animals in the previous version of the manuscript we agree and have changed the figure to reflect the entire testes (Figure 3A, B). We did not observe apoptosis in *Rlim*KO testes as measured via IHC using act. caspase 3 antibodies and mentioned this in the text (Discussion).

4) Figure 3C-E: it is unclear whether gene dysregulation in the KO occurs in germ cells or other cell types. The authors could sort different stages of spermatogenic cells in testis (example shown in Bastos et al., Cytometry Part A, 2005) and perform RNA-seq analysis.

We have targeted the cKO of *Rlim* to the early spermatogenic cell lineage or to Sertoli (and Leydig) cells using Ngn3-Cre and Sf1-Cre, respectively. While Ngn3-Cre *Rlim* cKO mice largely recapitulate defects of the systemic *Rlim* KO, SF1-Cre cKO mice do not (see also comments to point 6, below). Thus, much of the gene dysregulation in the KO likely occurs in the spermatogenic cell lineage.

5) Figure 3—figure supplement 1: the macro-H2A1 data is not convincing, and looks like background staining. I would suggest performing immunofluorescence staining of meiocyte nuclear spreads. Also, marker protein expression doesn't reveal whether XY silencing is affected. The authors' RNA-seq data could be used to ask whether sex-genes are de-repressed or not. Again, cell-type specific analysis suggested in comment 3 is important as the authors can focus the analysis on pachytene cell population, in which sex chromosome silencing occurs.

We agree with this criticism. Because *Rlim* exerts its function in the spermatogenic cell lineage, where it is highly expressed in round spermatids only after MSCI and meiosis has already occurred, it likely is not relevant for these processes and we have therefore removed this figure (see also point 3, reviewer 2).

6) Figure 4-6: the relationship between RLIM expression and the spermiogenesis phenotype is not clear. This mechanistic deficiency could be addressed using Cre lines that deplete Rlim in specific cell types (e.g. pachytene, Sertoli cells).

We agree with this criticism and have therefore targeted the cKO of *Rlim* to the spermatogenic cell lineage or to Sertoli (and Leydig) cells using Ngn3-Cre and Sf1-Cre, respectively. While Ngn3-Cre *Rlim* cKO mice largely recapitulate defects of the systemic *Rlim* KO, SF1-Cre cKO mice do not (Figure 7; see also comments to 4). Because *Rlim* is highly upregulated in round spermatids (Figure 2), this suggests major functions of *Rlim* during spermiogenesis. Together with the finding of defective cytoplasmic reduction (Figures 5-7), we believe the results shown in the revised manuscript clarify the relation between *Rlim* expression and the spermiogenesis phenotype.

Reviewer #2:This study reported a role of Rlim in male reproduction. RLIM is a RING finger ubiquitin E3 ligase. Previous studies have shown that Rlim knockout females are embryonic lethal due to a failure in the maintenance of X-inactivation in extraembryonic tissues but KO males are fertile. In the current study, Rlim expression was examined in both testis and epididymis at both protein and RNA levels. It was concluded that RLIM protein level peaks in pachytene spermatocytes. The Rlim KO males show reduced sperm count and decreased testis weight. The KO sperm display reduced motility and reduced fertilization rate in vitro. RNA-seq on adult KO and control testes revealed altered transcriptome. While sperm and fertilization defects are well characterized and interesting, the overall study is very preliminary. It is unclear how inactivation of Rlim causes such sperm defects. No attempt was made to identify the RLIM target proteins in testis. The fact that the Rlim KO males are still fertile diminishes its significance. There is a major problem with data interpretation. Overall, I don't feel that this preliminary study represents a significant contribution.1) Expression of RLIM protein in testis: The expression data interpretation (Figure 1) is not correct. It was concluded that RLIM expression peaks in pachytene spermatocytes at stages VI/VII (Figure 1B). RLIM-positive cells are also PNA-positive (acrosomal caps) and thus are round spermatids instead of pachytene spermatocytes. RLIM-positive round spermatids are obvious in the Figure 1A-1 tubule. Based on the data presented, the correct interpretation is that Rlim is expressed in round spermatids and Sertoli cells in the testis. While this mistake can be easily corrected, it has impacted the entire manuscript in a wrong way, which needs to be re-interpreted and re-written.

We agree with this reviewer and have addressed this issue, including re-interpretation and rewriting. Thank you for pointing this out.

2) Figure 2 shows the generation of a short testis-specific transcript due to alternative polyadenylation. This is quite common in testis and is not really meaningful.

While we agree that testis-specific transcripts via alternative polyadenylation are common in testis, we disagree that this is not meaningful, as it might help in the observed upregulation of *Rlim* protein observed in round spermatids (Figure 2, see also Discussion section), where *Rlim* actively regulates spermiogenesis. Moreover, the use of this polyadenylation signal is also observed during mouse preimplantation development, when *Rlim* exerts its crucial role in promoting X chromosome inactivation in females (see Wang et al., 2016; Figure 1E). Finally, our results identify a new and functional polyadenylation signal in the *Rlim* gene.

3) Impact on MSCI. First, Rlim is not expressed in pachytene spermatocytes (see point #1), thus discussion of MSCI is not relevant. Second, gammaH2AX and macroH2A1 are very abundant, their presence may not necessarily indicate lack of effect on MSCI. It is better to look at the expression of sex-linked (X-linked and Y-linked) genes from their RNA-seq data. If MSCI is not affected, sex-linked genes should not be preferentially affected.

Thank you for pointing this out. Based on this comment we have removed this figure (see also point 5, reviewer 1).

4) The RNA-seq was performed on adult whole testis instead of enriched germ cell types. Given that the cKO testis is smaller and the sperm count is lower, differential expression of genes (DEGs) identified could be due to the reduced proportion of germ cells in the KO testis. This needs to be taken into consideration.

Because the vast majority of cells in the testis reflect spermatogenic cells, where *Rlim* exerts its functions (Figure 7), we believe that differentially expressed genes as detected via RNA-seq (Figures 3D-F), likely reflect those expressed in this cell lineage, even though in the KO testis there is a slightly lower proportion of this cell type. We have addressed this point in the Discussion section.

Reviewer #3:Wang et al. identified the localization of RLIM in mouse testes and epididymis. They also made RlimcKO/Y mice and found decreased sperm production with high rates of morphological abnormalities as well as decrease in motility and fertilization rates in vitro. This study uncovered functions of RLIM in male reproduction, which may indicate potential evolutionary pressure for this gene. However, the study did not well illustrate the mechanism of the gene's functions in spermatogenesis, and some results they showed are not well analyzed. Thus, I suggest this paper to be rejected.1) Though the authors showed abnormal sperm morphology and decreased sperm motility in RlimcKO/Y mice, they did not identify the exact stage for the appearance of the phenotype, e.g. meiosis, post meiosis or during fertilization. Thus, they did not know the functional stage of this gene, which made it impossible for mechanism studies.

We have now shown that deletion of *Rlim* in the spermatogenic cell lineage is largely responsible for the sperm phenotype (Figure 7). Taken together with *Rlim* expression pattern in this lineage (Figure 2), our results provide strong evidence that *Rlim* in round spermatids is required for normal spermiogenesis (see also point 6, reviewer 1).

2) The authors suggested that RLIM has "high and dynamic expression in specific cell types including Sertoli cells as well as differentiating spermatocytes at the pachytene stage". However, according to the staining pattern of RLIM in Figure 1B, RLIM is exclusively localized in germ cells expressing PNA, which is only expressed in post meiotic cells. Thus, RLIM is not expressed in "differentiating spermatocytes at the pachytene stage".

Thank you for pointing this out (see also point 1, reviewer 2). We have corrected this error.

3) Most of the major conclusions raised in this study are supported by a single experiment, which are not very convincing.

It is unclear what the reviewer means with this comment. All experiments have been carried out several times on multiple biological replicates to obtain significant results. The testis phenotype is observed in mice with three different genetic settings: Germline *Rlim* KO, cKO/Y^Sox2-Cre^ and cKO/Y^Ngn3-Cre^ but not in cKO/Y^Sf1-Cre^ males (Figure 3—figure supplement 1, Figure 3 and Figure 7, respectively).

[Editors’ note: what follows is the authors’ response to the second round of review.]

Essential revisions:1) Reviewer 1, point 1) The authors showed that germline Rlim KO phenocopies the Sox2-Cre Rlim KO phenotype in sperm (Figure 7). In this experiment, quantification of RLIM protein decrease in the target cell types (germ cells in Ngn3-Cre KO and Sertoli cells in Sf1-Cre KO) should be shown.

Based on this comment, we now show that *RLIM* protein is undetectable in testes of *Sox2*-Cre cKO/Y mice via Western blotting (new Figure 2—figure supplement 1B), corroborating IHC results in Figure 2A. We have also performed IHC on testes sections of Ngn3-Cre cKO/Y and Sf1-Cre cKO/Y mice following *RLIM* expression. While Ngn3-Cre cKO mice lack immunoreactivity in the spermatogenic cell lineage, the staining of Sertoli cells appears normal. For Sf1-Cre cKO it is the inverse: immunoreactivity in Sertoli cells is low/not detectable while round spermatids stain normally. These results show correct targeting with high specificity and penetrance (new Figure 7A). See also point 10, below.

Also, is the IVF phenotype (Figure 4 F-G) also observed in the Ngn3-Cre Rlim KO? This will provide more important and direct evidence than the morphology analyses in Figure 7 to show sperm defects in the Ngn3-Cre Rlim KO.

Based on this comment we now show a similar IVF phenotype in the Ngn3-Cre cKO (new Figure 7G), thereby corroborating functions of *Rlim* in the spermatogenic cell lineage.

2) Reviewer 1, point 2) Figure 2A: The data requested was RLIM western blot using wildtype tissues to compare the RLIM protein size of the testis-specific short variant and the long variant expressed in other tissues. Because ORF is shared in these transcripts, their protein size would be same. This data should be added in the revision.

We have performed the requested Western blot (new Figure 2—figure supplement 1A) demonstrating expression of full length protein in testes.

3) Reviewer 1, point 9) It is stated that "the Cauda region of many cKO/YSox2-Cre animals contained thinner tubules when compared to control males". The requested data was quantification of this thinner tubule phenotype. This data should be added in the revision.

Based on this comment we have tried to quantify the thinner tubule phenotype using ImageJ. However, our results did not reach significance levels (P=0.14). Because 1. data on epididymis caused confusion (see points 12c, 13a), and 2. the *Rlim* KO phenotype is testicular, the epididymis is not of high relevance for the paper, we have removed this statement. All data on epididymis are now shown in a single Figure (Figure 3—figure supplement 2; see also points 12c and 13a).

Reviewer 1, point 10) Figure 4—figure supplement 1F suggests that Rlim deletion in sperm affects embryonic development after fertilisation. This point should be discussed more.

Because male mice lacking *Rlim* develop normally during preimplantation development (see Shin et al., 2010; 2014; see also Figure 3 —figure supplement 1A), our results in IVF (Figure 4—figure supplement 1F) indicates that the lack of *Rlim* appears to affect early embryo development specifically under in vitro conditions. We have added this in the text.

4) Throughout the Introduction including on the first line there is in appropriate reference to adult spermatogenesis involving primordial germ cells. These are not present in adult testes – should be undifferentiated spermatogonia or spermatogonial stem cells instead e.g. the progression of PGCs to mature spermatozoa should be progression of spermatogonia; stem cells to mature spermatozoa

Thank you for pointing this out. We have corrected this.

5) First paragraph of Introduction – midpiece is a section of the sperm tail so should read “formation of the sperm head and tail” or to be more precise “the formation of the sperm head, HTCA and tail” not “formation of the sperm head, midpiece and tail”

We have corrected this.

6) First paragraph of Introduction – “the number of genes involved remain limited.” Should be “the number of genes identified to be involved remains limited.”

Corrected.

7) When describing which spermatid population RLIM is expressed in should use spermatid step not just seminiferous tubule stage (as some tubules have two types of spermatids in them so it is better practice) e.g. the spermatids that it is expressing in stage VI/VII – are step 6-7 spermatids. Also, the meaning would be clearer especially to non-specialists if you change the following sentence:“RLIM levels are dramatically upregulated specifically in round spermatids that have undergone meiosis at stages VI / VII, an early timepoint in spermiogenesis (O'Donnell, 2014; Qian et al., 2014).”to“RLIM levels are dramatically upregulated specifically in post-meiotic step 6-7 round spermatids (stages VI / VII), an early timepoint in spermiogenesis (O'Donnell, 2014; Qian et al., 2014).”As previous wording could be read to mean that round spermatids undergo meiosis at stage VI / VII which is of course incorrect. Also step 6-7 isn't super early in spermiogenesis it's about 1/3-1/2 to half the way through, what is notable is it is immediately before spermatids begin to elongate in step 8. While you make reference to the fact that its expression is no longer detected in spermatozoa that are released during spermiation – it would also be good to make reference in the results text to the fact that its expression in other spermatid steps is low e.g. once they begin to elongate it goes down and beforehand in step 3 it is low. More precise definition of the all the spermatids steps it is upregulated in would be beneficial

Thank you for pointing this out. In response, we have rewritten this paragraph, addressing these points in the revised manuscript. We have also added a panel in Figure 2B that allows more precise staging of *RLIM* expression in spermatids. See also our response points 8 and 12b below.

8) Figure 1B bottom panel is incorrectly staged. Based on acrosome morphology that is likely stage XI not IX, as the acrosome morphology shows that the apical hook/falciform shape has been acquired

We agree and have adjusted the staging in Figure 2B (see also point 12).

9) More care needs to be taken with basic fertility phenotyping and characterisation of spermatogenesis. For example:a) The use of swim out from the epididymis is not a quantitative method of sperm numbers as it is dependent on the cuts made as to how many sperm vacate, in addition non-motile sperm will be less likely to vacate the epididymis. Accurate sperm counts require homogenisation of the epididymis, particularly given the KO mice have lower motility the intrinsic bias of the sperm swim out towards motile sperm may results in KO sperm counts appearing to be lower than they really are

We agree with this statement. We have changed the text to reflect for the intrinsic bias in swim out sperm numbers towards motile sperm. All sperm preps have been made by the same person (F.W.) ensuring similar cuts and the yields between animals of the same genotype were relatively reproducible (see error margins in Figure 3C).

b) Basic PAS analysis of testis histology of KO testes should be conducted, with particular attention to stage 8 and 9 to see if there are defects in spermiation i.e. are all sperm properly released or do you see spermatid retention at stage 9, do residual bodies form normally, normal size etc (this would help define whether the origin of the cytoplasmic droplet is due to defects during spermiation or if it is related to defects in the normal changes that occur to it during transit through the epididymis). In addition of the steps within spermatids undergo head shaping to identify the origin of these defects.

Based on this comment we have performed PAS analyses. Focusing on stage 9 tubules, we did not see signs of improper sperm release (new Figure 3—figure supplement 1D). Because low contrast did not allow for a reliable identification and interrogation of small vesicles including residual bodies, we examined signs of cytoplasmic reduction during spermiation in TEM on testis sections. We did not observe obvious/major defects during cytoplasmic reduction in TEM (new Figure 7—figure supplement 1D). However, as we were unable to quantify droplet/lobe size on spermatids in TEM because of differing section angles combined with an uneven distribution of the droplet over the midpiece, these results do not exclude mild defects. Incorporating these data, because of increased metabolite content (Figure 4A-C), our combined results suggest some defects in cytoplasmic reduction in *Rlim*KO spermatids during spermiation that might become exacerbated after sperm release due to difficulties in adjusting to a changing environment. This is discussed in the text.

c) Despite what the previous reviewer 3 said, testis and epididymis weight should not be expressed relative to body weight

Testis weight is commonly expressed relative to body weight. In this paper all three Cre drivers used are in a different genetic background: While *Sox2*-Cre mice are in a congenic C57Bl/6 background, the Sf1-Cre and the Ngn3-Cre mice are in different mixed backgrounds and animals display different sizes and weights and swim-out sperm numbers in controls differ (see Figure 7). We feel that it is important under these circumstances to normalize organ weights against the body weight to render data better comparable.

d) The double axoneme phenotype is not commonly seen in KO models and the origins of this might reveal important insight about sperm tail formation. Have the authors characterized the head to tail coupling apparatus of these sperm and investigated if this is due to supernumerary basal bodies?

The occurrence of sperm with 2 axonemes in *Rlim*KO animals is relatively rare (Figure 6B) and thus represents a minor phenotype. Staining swim-out sperm with antibodies against Centrin2 did not reveal frequent occurrence of supernumerary basal bodies. Therefore, we believe that the proposed investigations are beyond the scope of our study.

10) The authors need to include data showing validation of their KO models at an mRNA and protein level

See also our response to point 1. Using *Sox2*-Cre to target the *Rlim* cKO, we have previously demonstrated correct targeting of the floxed region at the mRNA level via single embryo RNA-seq (see Wang et al., 2016; Figure 1E; the parental *Rlim* has been targeted). Correct and efficient targeting of *RLIM* protein in testis by *Sox2*-Cre is now shown via IHC (Figure 2A) and Western blot (Figure 2—figure supplement 1B). Concerning Ngn3-Cre cKO and Sf1-Cre cKO mice, we have performed IHC on testes sections showing correct targeting with high specificity and penetrance (new Figure 7A).

11) How do the authors think the gene transcription changes in the RLIM KO mice relates to the phenotypes observed? Some interpretation of this is needed in the Discussion

Based on this comment we have added a paragraph on the interpretation of our RNA-seq results in the Discussion section, as requested.

12) Result section Rlim expression in testis is highly regulateda) It is not clearly presented when exactly RLIM is highly expressed in postmeiotic cells. Is it restricted to stages VI to VIII? Since at stage IX on Figure 2B the signal is no longer strong (but see my comment below regarding staging). On Figure 2A, it seems as if many tubules display strong round spermatid signal. A precise description of RLIM protein dynamic at various stages of spermatogenesis is needed. Recapitulating IF/IHC observations using a scheme showing the different stages of spermatogenesis would be useful.

We have added a new panel in Figure 2B that shows high *RLIM* expression in step 7-8 spermatids. Thus, *RLIM* expression is high in spermatids from step 6-8, but low at spermatid stages before and after that. We have made this clear in the text. See also our response to point 7.

b) Figure 2A – IHC panel:Higher magnification insets from negative CTL would be useful to confirm that the signal observed in Sertoli cells is specific (i.e. not seen in neg control)

Based on this comment we have added a higher magnification inset in Figure 2A, which we believe makes it clear that *RLIM* is no longer present in Sertoli cells in the *Sox2*-Cre cKO/Y.

B – To me (but the figure is small) the image labelled "stage IX" is not a stage IX. I do not recognize the lectin PNA signal to be a stage IX: At this stage round spermatids have just started to elongate and condensed step 16 spermatids have already undergone spermiation (so are no longer present).NB. Higher magnification of the DAPI images (in the merged figure for instance) would be helpful.

We agree and have changed the Figure accordingly and added a higher magnification inset of DAPI images, as requested. See also our response to points 7 and 12 above.

c) It seems odd to present Rlim expression in epididymal cells in the part entitled "Rlim expression in testis is highly regulated"

In response to this comment and points 3 and 13a, to avoid confusion, we have summarized all data on epididymis in a single paragraph and figure (Figure 3—figure supplement 2), as *Rlim* does not play a major role in epididymis.

13) Result section Diminished production and functionality of Rlim KO sperma) Overall I find this part a bit confusing. At first, the testicular impairment seems to be minimized because of the observation that Rlim is expressed in epididymis epithelium. Yet it clearly appears from the last result section that sperm defects originate from a testicular (germ cell) impairment. Maybe this result section could be slightly re-organized?specific comments/questions:

In response to this comment and points 3 and 12c, to avoid confusion, we have summarized all data on epididymis in a single paragraph and figure (Figure 3—figure supplement 2), as *Rlim* does not play a major role in epididymis

b) The observation of a small but significant reduction in testis weight indicates a (mild) spermatogenesis defect; so do testicular sperm morphological abnormalities.Can the authors further investigate testicular cell composition – to see if one particular (likely postmeiotic) cell stage is affected? In the Discussion, it is said that increased cell death and chromatin packaging defects were investigated – even if the results are negative it would be worth presenting (or evoking) them in this section.

Our PAS analyses (Figure 3—figure supplement 1D; data not shown) did not reveal major visible changes at the level of testicular cell composition. We have mentioned in the Results section that no cell death was observed. No obvious chromatin packaging defect was observed as judged by sperm head density in TEM and is mentioned in the TEM section.

Since spermatogenesis is mildly impaired, RNAseq on whole testis should be interpreted with caution, as it could reflect a mild change in cell population. This emphasizes the need for a more detailed presentation of the testicular defects (or absence of visible defects).

We agree with the reviewer. We have added in the Discussion section the sentence: “However, we cannot exclude the possibility that subtle and hence undetected changes in testicular cell types in mice lacking *Rlim* (Figure 3—figure supplement 1D) may contribute to the observed differences in gene expression as well as testes weight (Figure 3).”

c) Sentence: "…revealed that the Cauda region of many cKO/YSox2-Cre animals contained thinner tubules when compared to control males (Figure 3—figure supplement 1E), consistent with decreased Caudal sperm (Figure 3C)."This sentence suggests that decreased caudal sperm is due to caudal region malformation. But to conclude, one needs to show quantification of testicular spermatozoa (or caput spermatozoa). Figure 6A data could be presented at that point.But then I am confused: what is the authors' conclusion? is the decreased sperm count a consequence of testicular or epididymis defects? Since epididymis defects appear to be ruled out by the analysis of Ngn3-Cre KO sperm, it would increase clarity if epididymis analyses were presented separately.

See also our response to points 3 and 12c. Based on this comment we have we have summarized all data on epididymis in a single paragraph and figure (Figure 3—figure supplement 2), as requested.

14) Discussiona) The first part about Rlim short isoform could be reduced. While it is an interesting fact, I feel it does not require so much emphasis since testis specific isoforms are quite common.

We agree and have condensed this section in the Discussion.

b) The fact that Rlim is X-encoded and expressed in spermatids does not mean it escapes MSCI (as MSCI occurs in pachytene primary spermatocytes). Instead, it means that Rlim short isoform is one of the many X-linked genes that are expressed after meiosis, when the X is re-activated (see for instance Ernst et al., 2019)

Thank you for pointing this out. This has been corrected in the revised manuscript.

c) Considering the fact that the KO impact on male reproductive abilities (at least for the lab mouse) is minor, I find the part about the evolutionary consequences of a spermatogenesis role for Rlim a bit far-fetched. The paternal effect of Rlim in milk-producing alveolar cells would likely be more predominant (Jiao et al., 2012). I would recommend to put less emphasis on that aspect, especially in the Abstract.

We agree and have removed this aspect in the Abstract and shortened the corresponding paragraph in the Discussion.